# Pancreatic Ductal Adenocarcinoma (PDAC) circulating tumor cells influence myeloid cell differentiation to support their survival and immunoresistance in portal vein circulation

**Juan Pablo Arnoletti[1‡], Joseph Reza[2], Armando Rosales[1], Alberto Monreal[2], Na'im Fanaian[3], Suzanne Whisner[4], Milan Srivastava[4], Julia Rivera-Otero[4], Gongxin Yu[4], Otto Phanstiel IV[5], Deborah A. Altomare[6], Quang Tran[7], Sally A. Litherland[7‡]***

1 Center for Surgical Oncology, AdventHealth Cancer Institute, Orlando, Florida, United States of America, 2 General SurgeryResidency Program, AdventHealth, Orlando, Florida, United States of America, 3 Central Florida Pathology Associates, Orlando, Florida, United States of America, 4 AdventHealth Research Institute, Orlando, Florida, United States of America, 5 Department of Medical Education, College of Medicine, University of Central Florida, Orlando, Florida, United States of America, 6 Burnett School of Biomedical Sciences, College of Medicine, University of Central Florida, Orlando, Florida, United States of America, 7 Translational Research, AdventHealth Cancer Institute, Winter Park, Florida, United States of America

‡ JPA and SAL are co-senior authors on this work.
* sally.litherland@adventhealth.com

**Data Availability Statement:** All relevant data are within the manuscript and its Supporting Information files. Minimal data sets have been

## Abstract

The portal venous circulation provides a conduit for pancreatic ductal adenocarcinoma (PDAC) tumor cells to the liver parenchyma sinusoids, a frequent site of metastasis. Turbulent flow in the portal circulation promotes retention of PDAC shed circulating tumor cells (CTC) and myeloid-derived immunosuppressor cells (MDSC). Excessive colony stimulating factor-1 receptor (CSF1R) signaling can induce myeloid differentiation to MDSC and transformation of MDSC to myeloid-derived fibroblasts (M-FB). Interactions between PDAC CTC and M-FB in the portal blood promotes the formation of immunoresistant clusters that enhance CTC proliferation, migration, and survival. Analysis of portal and peripheral blood samples collected intraoperatively from 30 PDAC patients undergoing pancreatico-duodenectomy showed that PDAC patient plasma contained high levels of macrophage colony stimulating factor (M-CSF/CSF1), granulocyte-macrophage colony stimulating factor (GM-CSF/CSF2), interleukin-8 (IL-8), and interleukin-34 (IL-34) compared to healthy control levels. Moreover, the level of M-CSF in portal blood was significantly higher than that detected in the peripheral blood of PDAC patients. PDAC CTC aseptically isolated by fluorescence activated cell sorting (FACS) out of freshly collected patient portal blood mononuclear cells (PortalBMC) had elevated RNA expression of *IL34* (IL-34 gene) and *CSF1* (M-CSF/CSF1 gene) which both signal through CSF1R. PDAC CTC also had high levels of RNA expression for *CXCL8*, the gene encoding chemokine interleukin-8 (IL-8) which can attract myeloid cells through their CXCR2 receptors. FACS-isolated portal PDAC CTC and M-FB co-cultured *ex vivo* had increased CTC proliferation, motility, and cluster formation compared to CTC cultured alone. CSF1R and CXCR2 cell surface expression were found on PDAC

uploaded in Supporting Information files with links to GeneWiz data for FigS2 included. Additional methods and supportive materials are given in previous publications: Arnoletti et al 2017/ Pancreas. 2017 Jan;46(1):116-123. doi: 10.1097/ MPA.0000000000000667 and Arnoletti et al2018/ CBT Cancer Biol Ther. 2018;19(10):887-897. doi: 10.1080/15384047.2018.1480292. Epub 2018 Aug 1.

**Funding:** Research funding was provided by grants from Bristol Myers Squibb (BMS CA025-016, SAL; bms.fasttrack.com), Phi Beta Psi Philanthropic Sorority (SAL/JPA;www.phibetapsi.org), Fraternal Order of Eagles President's Charity (JPA/SAL, SAL/ JPA; https://clicktime.symantec.com/ 35XhzGkkVbmaV4i997mRESa6xU?u=www.foe. com), State of Florida Health Department Bankhead-Coley Research (OP/JPA/SAL; floridahealth.gov/research) and AdventHealth Foundation philanthropic gifts (adventhealth.com/ adventhealthfoundation). The funders had no role in the study design, data collection and analysis, decision to publish or preparation of the manuscript.

**Competing interests:** Bristol Myers Squibb was the supplier of the humanized antibodies and a research funding source used for our studies. We can confirm that this does not alter our adherence to the PLOS One policies on sharing data and materials.

portal blood CTC and M-FB, suggesting that both cell types may respond to M-CSF, IL-34, and IL-8-mediated signaling. Portal PDAC CTC displayed enhanced RNA expression of *CSF1* and *IL34*, while CTC+M-FB+ clusters formed *in vivo* had increased RNA expression of *CSF2* and *IL34*. Portal M-FB were found to have high *CSF1R* RNA expression. CTC isolated from e*x vivo* 7-day cultures of PDAC patient portal blood mononuclear cells (PortalBMC) expressed elevated *CSF1*, *IL34*, and *IL8* RNA, and *CSF1* expression was elevated in M-FB. Treatment with rabbit anti-CSF1R antibodies decreased CTC proliferation. Treatment of PortalBMC cultures with humanized anti-CSF1R, humanized anti-IL-8, or anti-IL-34 antibodies disrupted CTC cluster formation and increased CTC apoptosis. U937 myeloid precursor cell line cultures treated with conditioned media from PortalBMC *ex vivo* cultures without treatment or treated with anti-IL-8 and/or anti-CSF1R did not prevent myeloid differentiation in the myeloid precursor cell line U937 to macrophage, dendritic cell, MDSC, and M-FB phenotypes; whereas, U937 cultures treated with conditioned media from PortalBMC *ex vivo* cultures exposed to anti-IL-34 were significantly inhibited in their myeloid differentiation to all but the M-FB phenotype. PDAC patient T cells that were found phenotypically anergic (CD3+CD25+CTLA4+PD1L1+) in PortalBMC could be re-activated (CD3+CD25 +CTLA4-PD1L1-), and displayed increased interferon gamma (IFNγ) production when PortalBMC *ex vivo* cultures were treated with anti-CSF1R, anti-IL-8, and anti-IL-34 antibodies alone or in combination. These findings suggest that PDAC CTC have the potential to influence myeloid differentiation and/or antigen presenting cell activation in the PDAC portal blood microenvironment, and that disruption of CTC/M-FB interactions may be potential targets for reversing the immunosuppression supporting CTC survival in the portal blood.

## Introduction

Pancreatic ductal adenocarcinomas (PDAC) are among the most morbid human cancers [1]. Their deep anatomic location and resilience in the naturally immunosuppressed environs of the gastrointestinal system make these tumors difficult to detect early and treat effectively [2]. As a result, systemic chemotherapeutic regimens have very modest efficacy and T cell directed immunotherapies that have proven beneficial in other aggressive tumors such as lung, melanoma, renal, and breast cancers, have been shown to be ineffective in patients with PDAC [1, 3, 4]. Surgical resection is currently the only potentially curative treatment, but only if applied early and expertly [3, 5]. Unfortunately, less than 20% of all PDAC patients are candidates for surgical resection with curative intent at diagnosis [3–5].

A major contributor to PDAC-related mortality is the propensity for rapid development of distant metastasis [1]. In PDAC progression, tumor cells released from primary tumors enter the portal venous blood, a circulatory compartment with low-pressure turbulent and bi-directional flow, allowing for retention of cells away from general circulation and filtration by the liver [6, 7]. Tumor cells shed from primary PDAC tumors can form aggregate clusters that may promote tumor survival in the portal venous blood and beyond, contributing to tumor progression after surgical resection [8, 9]. Understanding how these circulating cell subpopulations circumvent the host immune response to survive and enable tumor growth in metastatic sites is critical to finding effective systemic and targeted treatment strategies.

In the portal blood, PDAC circulating tumor cells (CTC) can attract and influence myeloid precursor cells which can aid in CTC survival, motility, and extravasation to surrounding

tissues [7, 9–12]. Myeloid precursor cells are found normally in blood that can give rise to multiple cell types through differentiation driven by myeloid growth and inflammatory factors found in the portal circulation [13–16]. Myeloid cell growth and differentiation factors including macrophage colony stimulating factor (M-CSF/CSF1), granulocyte-macrophage colony stimulating factor (GM-CSF/CSF2), and interleukin-34 (IL-34) act on myeloid cells to promote their differentiation into antigen presenting cells such as dendritic cells, monocytes, and specialized B cell-derived macrophages [13, 14]. Inflammatory factors such as prostanoids and TNFα can drive myeloid cells toward functions as aggressive phagocytic killing cells [15, 16]. Tumor cells that can produce these same factors may be able to alter the signaling effects on myeloid cells in their microenvironment, possibly by using different localized concentrations or timing [13, 14, 17].

Portal blood myeloid cells in contact with prolonged exposure to M-CSF or tumor cells can be influenced to differentiate toward fibroblast-like phenotypes with tumor support and immunosuppressive functions. These myeloid-derived suppressor cells (MDSC) and their derivative myeloid-derived fibroblasts (M-FB) can act as shields against immune attack, source of factors, provide metabolic support, motility enhancement for extravasation, and scaffolding for metastatic establishment once out into surrounding tissue [10–12, 18, 19]. Therefore, blocking CTC interactions with myeloid cells may decrease functions key to metastatic re-emergence including CTC proliferation, resistance to apoptosis, and motility [10–12, 20].

In this study, we examined how portal blood PDAC CTC can attract and influence circulating myeloid cells into M-FB differentiation to aid in CTC survival, proliferation, and motility.

## Methods & materials

### Sample collection

Portal and peripheral venous blood samples were collected during pancreatico-duodenectomy performed with curative intent in 30 patients diagnosed with pancreatic head ductal adenocarcinoma (PDAC, 14 women, 16 men). Written patient informed consent was obtained prior to surgical treatment, and participation in the study was performed under AHCI IRB approved protocols 946704/507397/562565. Eligible patients for this study were adults ≥ 18yr of age diagnosed with localized PDAC who were deemed candidates for surgical treatment.

The median age for the patients at surgery was 62 years (range 48–77, mean 64.79± SD 9.01). The patient population was 13% Hispanic and 97% Caucasian. Twenty-one patients received chemotherapy prior to surgery (FOLFIRINOX, Capecitabine, or Gemcitabine/Abraxane). Eleven of these patients also received preoperative external beam radiation therapy. Patient tumor staging data were collected from post-surgical specimen pathological diagnosis after analyses were completed. Pathological staging included tumors of 9 patients staged as T1N0, 1 patient as T1N+, 6 patients as T2N0, 8 patients as T2N+, 3 patients as T3N0, and 3 patients as T3N+. With the exception of CTC numbers in portal blood as previously reported [7, 11], neither pre-surgical treatment or tumor stage were found to significantly impact the statistical analyses of outcomes in datasets where the subpopulations were compared or in the bioinformatics on pilot RNA SEQ data mentioned in these studies.

### FACS isolation of portal blood CTC and M-FB cell populations

CD44+ CD147+ EPCAM+/CK+ CTC [7, 11, 21, 22] and CD14+CD105+ M-FB [11] candidate cells were aseptically sorted individually or when found together in *in vivo*-formed clusters out of ficoll histopaque (Sigma-Aldrich) isolated PortalBMC using a 10-parameter/8-color Beckman Coulter MoFlo XDP FACS instrument fitted with a 100 micron nozzle [7, 11]. When enough cells for extended pre-culture sorting were available, both pan-Cytokeratin (CK) and

EPCAM biomarkers were used in tandem and both EPCAM+CK- and EPCAM+CK+ populations collected for further analysis together to enrich for potential EMT transition CTC populations. Antibody fluorescent conjugates used in FACS and in biomarker identification on the Beckman-Coulter MoFlo or a BD Biosciences CANTOII flow cytometer, were obtained from BD Biosciences (αEPCAM (347198, 347197), αpan Cytokeratin (550953), αCD105 (562408, 561443), αCD14 (340585), αCD147 (562551)), and from BioLegend αCD44 (338816). Antibody conjugates were standardly used at 1μg/million cells. 7AAD/AnnexinV (BD BioSciences Apoptosis Kit (559763)) staining for viable/apoptotic cells was detected by flow cytometric analysis. On average, we collected 35338.46±59742.90SD/ median 11932 CTC per million PortalBMC sorted directly from the PDAC patient samples prior to culture. The variability of CTC collection and plasma volumes available was high between patients and not all samples gave enough cells or plasma to complete all analyses. The n value for the subset of samples used in each experiment are listed in the Figure legends.

## Single cell sorting & RNA SEQ analyses

For single cell sorting (SCS) used in RNA expression analysis, the CyClone robotic adaptor on the MoFlo FACS instrument was calibrated to collect 1 CTC per PCR tube collected directly into RNALater (Ambion/ThermoFisher) in a 96-tube PCR plate format. Representative SCS CTC samples from 2 PDAC patients were shipped frozen in RNALater to GeneWiz (NJ) for RNA extraction, library production, and RNA SEQ analysis. Bioinformatic analysis of the resultant data was performed by GeneWiz and in-house by co-author Dr. Yu. Comparison of single cell RNA SEQ to oligo CTC sample RNA SEQ for 2 PDAC patients. We ran quantitative RT-PCR confirmatory analyses for enriched genes on additional 3 clone single cell CTC samples from 12 PDAC patients. These confirmatory analyses showed that there was clonal RNA expression variation within each patient's portal blood sample. These preliminary RNA SEQ data were used to select genes for RNA expression analysis RT-PCR panel pertaining to myeloid differentiation and CTC survival/growth functions used to survey PDAC patient RNA extracts of FACS isolated CTC and M-FB from uncultured patient PortalBMC, as well as from PortalBMC and defined mixed cell cluster *ex vivo* cultures.

## Defined FACS isolated mixed cell cultures for cluster essential cell components and motility studies

PortalBMC of 18 PDAC patients were subjected to pre-culture FACS isolation for separation of CTC (CD44+CD147+EPCAM+/CK+CD45-), MDSC (CD14+CD33+MHCII DR-), DC (CD11c+MHCII DR+CD14-), T cells (CD3+CD45+), and M-FB (CD14+CD105+MHCII DR-) as well as collection of negatively selected (i.e., -CTC, -M-FB, -MDSC, -DC, -Tcell) mixed populations as previously described [11]. Each isolated population was cultured alone or in mixed culture with CTC from the same patient immediately after FACS isolation in rich media (RPMI (Cellgro/Corning), 10% 199 medium (Cellgro/Gibco), 10% heat-inactivated, low endotoxin fetal bovine serum (Cellgro/Corning), and 2% penicillin-streptomycin- amphotericin B antibiotic-antimycotic mix (Sigma-Aldrich)). *Ex vivo* cultures were incubated at 37˚C/5%CO$_2$ for up to 10 days for analysis of CTC cluster formation, CTC proliferation detection by biomarker and CSFE flow cytometry, and RNA expression analysis. Cultures used for testing the effects of myeloid growth factor signaling on CTC clusters were treated with 5nM GM-CSF, 0.5μg/ml M-CSF, 1μg/million cells rabbit monoclonal anti-CSF1R antibodies (Abcam ab183316), or mouse polyclonal anti-CSF2R common chain (Abcam ab95681) for the duration of cluster culture.

We have found that CTC alone can be maintained in culture, cryogenetically preserved, and re-grown from thaw to form monolayer culture similar to seen in transformed, established PDAC cell lines. However, CTC clusters, once formed, do not survive or re-form after cryogenetic suspension and will progress to altered morphology if allowed to grow in continuous culture over time greater than 14 days, altering their *'ex vivo'* characteristics. To avoid *in vitro* transformation of the CTC, we limited the scope of our analyses to cells grown in culture up to 14 days.

## CTC motility & initiation of cluster formation studies

A portion of CTC and M-FB FACS isolates collected from uncultured PortalBMC were labeled as per the dye manufacturer's specifications with CSFE (Molecular Probes/ThermoFisher) and MitoTracker Red (Molecular Probes/ThermoFisher), respectively, and grown for 2 to 22hr at 37°C/5%CO$_2$ in transwell chambers (Sigma Aldrich) to detect cell migration potential alone and in co-culture, and for analysis of cell proliferation/identification by flow cytometry; and in chamberwell slides (Nunc/ThermoFisher) for cluster formation analysis by DIC microscopy.

Cell migration was analyzed by loading CSFE labeled CTC in 8µ pore membrane, fluorescent-blocking transwell inserts (Sigma Aldrich) with or without M-FB from the same patient, with the cells mixed at the same ratio as detected in FACS isolation from uncultured PortalBMC. CTC migration was calculated from the CSFE fluorescence detected in the lower well chamber within 24hr of culture using a BioTek fluorescence spectrophotometer plate reader.

## PortalBMC and defined CTC/M-FB cluster studies

For defined cell co-cultures, PortalBMC were subjected to FACS isolation to collect CTC, M-FB and clusters that were formed *in vivo* containing both CTC and M-FB. Cells remaining after FACS selection were also collected as 'negatively selected' (e.g. -CTC-M-FB) cell populations. Defined cultures of CTC alone, M-FB alone, *in vivo*-formed clusters, negatively selected cells, and unseparated PortalBMC were grown in rich media (RPMI,(Cellgro/Corning) 10% 199 medium, (Cellgro/Gibco) 10% heat-inactivated, low endotoxin fetal bovine serum (Cellgro/Corning) and 2% penicillin-streptomycin-amphotericin B antibiotic-antimycotic mix (Sigma-Aldrich)) at 37°C/5%CO$_2$ for up to 10 days and examined by DIC microscopic imaging using a Zeiss Axiovert fluorescence DIC microscope. Cluster formation in *ex vivo* cultures was measured by aggregate area measurements using NIH Image J software. CTC apoptosis, and proliferation were analyzed by Annexin V/7AAD and CTC biomarker flow cytometry.

Cultures used for testing the effects of myeloid growth factor signaling on CTC clusters were treated with 5nM GM-CSF, 0.5µg/ml M-CSF, 1µg/million cells (calculated to be approximately 0.07mg/kg, based on an estimated plasma volume of a 91kg patient), rabbit monoclonal anti-CSF1R antibodies, (Abcam ab183316), mouse monoclonal anti-IL34 (Abcam ab101443), or mouse polyclonal anti-CSF2R common chain (Abcam ab95681) during culture. After the successful findings of initial experiments with Abcam antibodies to anti-CSF1R, we tested Bristol Myers Squibb humanized IgG4 monoclonal anti-CSF1R antibodies to block CSF1R (2–8 mg/kg based on an estimated plasma volume of a 91kg patient and dosages used in Phase 1/2 clinical trials (FPA008-01, FPA008-002, CA025001, and CA025006; BMS Investigator Brochure BMS-986227), BMS-986227, Cabiralizumab), and Bristol Myers Squibb humanized IgG1 monoclonal antibodies to block the chemokine interleukin-8 (IL-8, 5-20mg/kg based on an estimated plasma volume of a 91kg patient and dosages used in Phase 1/2 clinical trials (CA027-001, HuMax-Inflam-001, Hx-Inflam-302, BMS Investigator Brochure BMC-986253), BMS-986253) in subsequent analyses.

## Phenotypic analysis of cells in *ex vivo* PortalBMC cluster cultures

After incubation, defined co-cultures of CTC/M-FB and PortalBMC *ex vivo* cultures were sorted by aseptic FACS into EPCAM+/CK+CD44+CD147+ CTC and CD14+CD105+ M-FB cell populations and analyzed for cell surface markers including CXCR2(BD Biosciences 551125) and CSF1R(BD Biosciences 565346) by flow cytometry. In addition, myeloid phenotypes within PortalBMC *ex vivo* cultures were identified by flow cytometry using CD14 +MHCII DR+ to identify monocytes/macrophages (Mφ), CD11c+MHCII DR+ for dendritic cells (DC), and CD14+CD33+ MHCII DR- CD11c- for myeloid-derived suppressor cells (MDSC). CD3+ T cells detected in unsorted PortalBMC cluster cultures were also analyzed by flow cytometry for expression of CTLA4, CD25, PD1, and PD1L1 to assess their anergic/activation states.

## Cytokine/Myeloid growth factor analysis

Plasma samples, collected after centrifugation from the top of the ficoll gradient separation of portal blood, were frozen at -80˚C for later use in cytokine/growth factor analyses. GM-CSF, M-CSF, IL-34, IL-8, interleukin 1alpha (IL1α), interleukin 1beta (IL1β), prostaglandin E2 (PGE2), and IFNγ levels were measured in portal and peripheral blood plasma and in media supernatants of PortalBMC cells grown in culture for cluster formation ('conditioned media') using BioLegend Legendplex multi-analyte assay flow cytometry or conventional ELISA (Boster Scientific, Cayman Chemical). Cell-free culture supernatants were collected by plate centrifugation of 7 day cultures at 600xg, 5min at 25˚C and frozen at -80˚C until analysis in duplicate by ELISA. Since portal blood from healthy individuals is not accessible without invasive intervention, portal blood control samples could not be ethically obtained for comparison. Therefore, portal and peripheral blood samples were compared to manufacturer/literature cited mean or median values for healthy human peripheral blood plasma (mean 33.7pg/ml GM-CSF, BioLegend, mean 83.7pg/ml M-CSF, BioLegend, mean 7.13pg/ml IL-34 [23], median 83pg/ml IL-8 [24], mean 1.91pg/ml IFNγ [25], mean 0.74pg/ml IL1α [26], mean 1.85pg/ml IL1β [27], mean 312pg/ml PGE2, ThermoFisher).

## U937 precursor myeloid cell line differentiation assays

Plasma and *ex vivo* PortalBMC culture conditioned media were added at 25% media volume to counted cell number cultures of the myeloid precursor cell line U937 (ATCC) to examine the potential for PDAC blood soluble growth factors to affect myeloid differentiation. After 7 days at 37˚C/5%CO$_2$ in culture, the U937 cells were harvested with cell dissociation media (Sigma-Aldrich) at 4˚C, 15-30min, and analyzed by flow cytometry for biomarkers of development of myeloid differentiation/ antigen presenting cell (APC) markers relative to those found on untreated U937 cells grown without conditioned media: CD14+ (BD Biosciences 340585)/ MHCII DR+ (BD Biosciences 560743) for monocyte/macrophage APC, CD11c (BD Biosciences (561355)/MHCII DR+ for dendritic cell APC, and CD14+CD33+CD11c-DR- for myeloid-derived suppressor cells (MDSC, anti-CD33, BD BioSciences (551378), and CD14 +CD105+ for myeloid-derived fibroblasts (M-FB, anti-CD105 BD Biosciences (562408, 561443)).

## RNA analysis

Portal blood single cell FACS isolated clones (SCS) of CTC and M-FB, and cluster isolates from pre- and post- *ex vivo* culture FACS isolation were suspended in 1 to 1 mix of 1x PBS/ RNALater (Ambion/ ThermoFisher) and stored at -80˚C for later analysis. RNA was extracted

from thawed cells using Invitrogen (ThermoFisher) or Sigma-Aldrich (Millipore-Sigma) low cell RNA extraction kits, then quantified and purity checked (260/280 ratio) using a BioTek microdrop spectrophotometer.

RNA was stored at -80˚C until analyzed by real time RT-PCR using primers for *KRAS* exon 12 mutation (*KRASmut*, F5′-ACC TTATGT GTG ACATGT TCTAATATA GT-3′ and R5′-GCA CTC TTG CCT ACG CGA T-3R′, with probe FAM 5′-CCT GCTGAAAATGACTGAA-TATAAACTTGTGG-MGB for exon 12 12Ala, 12Arg, 12Asp, 12Cyc, 12Ser, and 12Val mutations, IDT), M-CSF receptor *CSF1R* (Hs00911250_m1, ThermoFisher), GM-CSF receptor *CSF2R* (Hs 00531296_g1, ThermoFisher), GM-CSF gene *CSF2* (Hs99999044_m1, Thermo-Fisher), IL-8 gene *CXCL8* (Hs00174103_m1, ThermoFisher), chemokine receptor *CXCR2* (Hs. PT.58.40527202, IDT), IL-34 gene *IL34* (Hs01050926_m1, ThermoFisher), oncogene *RAN* (Hs01590289_g1, ThermoFisher), osteopontin gene *SPP1* (Hs00959010_m1, ThermoFisher), proto-oncogene *RET1* (F5′ACA GGG GAT GCA GTA TCT GG and R5′CCT GGC TCC TCT TCA CGT AG, IDT) and control gene *GAPDH* (Hs 02758991_g1, ThermoFisher). RT-PCR was performed using Applied Biosystems One-Step RT-PCR reagents in 30–40 cycle protocols using ChaiPCR or Applied Biosystems MicroArray thermocyclers. RT-PCR results are reported as RNA copy number = $2^{(CtGAPDH-CtTarget\ gene)}$.

## Statistical analysis

Multi variant statistical analysis for each data set was done using GraphPad Prism version 5 or version 9 for ANOVA with Tukey or Bonferroni pairwise post testing or Kruskal Wallis non-parametric analysis with Dunns' pairwise post testing, using 95% confidence intervals and p<0.05 as significance threshold. Pairwise analyses were done with paired, unpaired, or single value compared to control analyses with Student t-testing or non-parametric Mann Whitney u testing where appropriate, using 95% confidence and p<0.05 as significance threshold. Sample size varied for data sets due to the patient to patient sample variation in CTC numbers and plasma volume limiting the number of analyses possible. Sample n values are listed in the Figure legends for each analyses. Unless otherwise indicated, lines and error bars on figures depict mean ± standard error.

GeneWiz bioinformatics analysis services were used for single cell RNA SEQ analyses. In-house bioinformatics analysis on SCS RNA SEQ followed a standard analysis procedure with minor modifications [28]. Briefly, RNA SEQ data were first converted to transcripts per million (TPM) [29], then $log_2$-transformed and z-scaled for data normalization and standardization. Limma (version 3.46.0) [30], an R/Bioconductor software package, was subsequently applied to identify differentially expressed genes (DEGs). Once (DEGs) were identified, clustering analyses (PCA and heatmaps) were performed with factoextra (version 1.0.7) and gplots (version 3.1.1), and two R packages [31]. The analyses were done with the 100 top-ranked DEGs by randomForest classification (R-package version of 4.6–14) [32] to assess the disease status. The functional analyses were performed with ClusterProfile [33, 34] and ReactomePA [35], and two Bioconductor packages to identify significantly enriched biological pathways.

## Results

### Portal CTC cluster formation requires M-FB cell populations

After 7-days *in ex vivo* culture, PortalBMC spontaneously formed aggregations of CTC with other surrounding cells that promoted CTC proliferation ([11] & **Fig 1**). When FACS-isolated portal CTC and immune cell types were placed in defined co-cultures, analysis of the resultant clusters indicated that though various cell types could participate in portal cluster formation

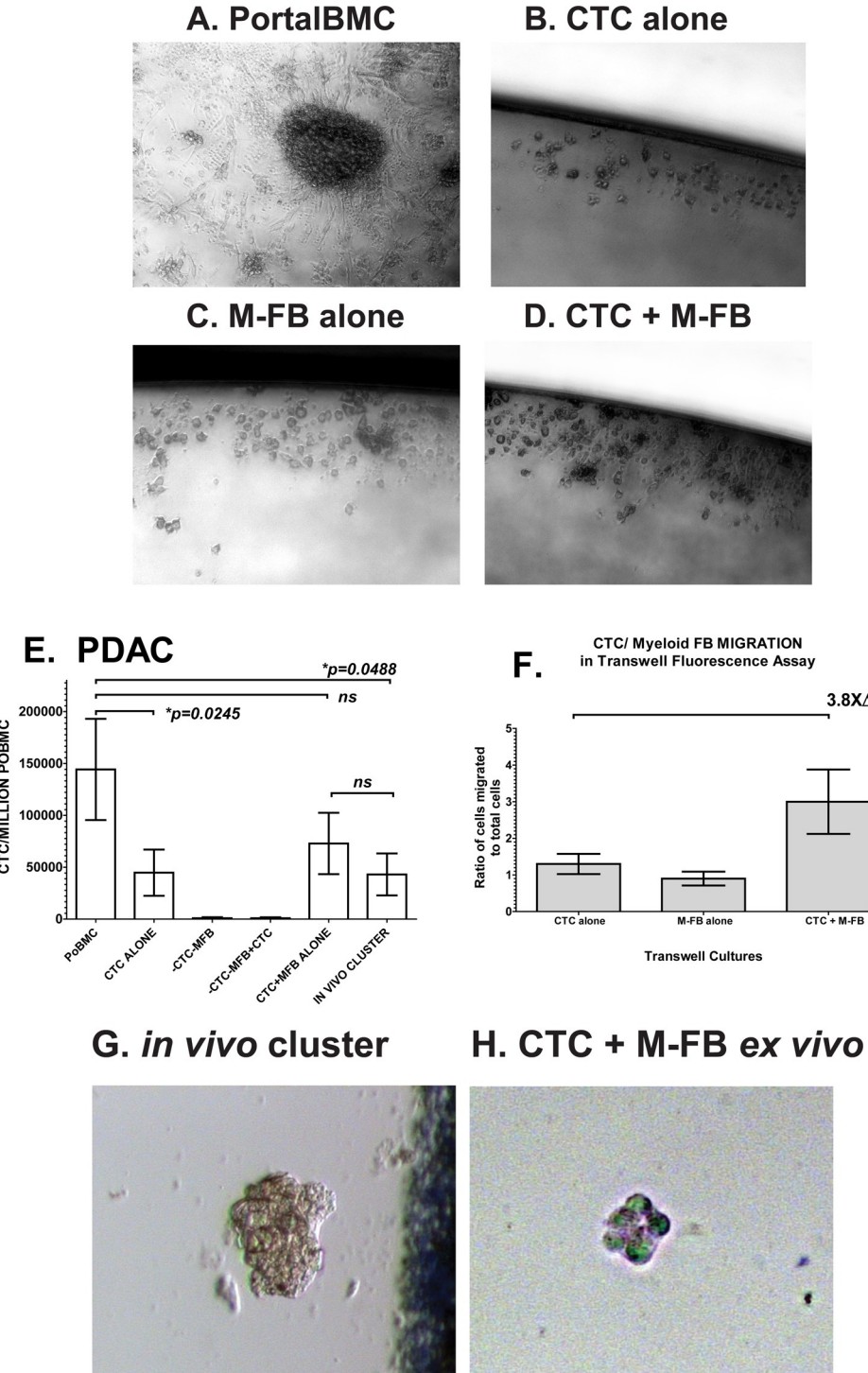

**Fig 1. PDAC CTC/M-FB clusters promoted CTC proliferation and migration.** PortalBMC were separated on histopaque gradients and CTC and M-FB cells isolated using aseptic FACS to yield selected populations for defined co-cultures to identify essential populations needed for CTC cluster formation in *ex vivo* cultures as previously described [11]. Clusters formed in 7-day cultures (**A,D**) by interactions between CTC and M-FB. Neither CTC (**B**) or M-FB (**C**) alone form clusters, though CTC alone can proliferate as a monolayer culture. CTC in 7 day *ex vivo* cultures of unsorted PortalBMC (**E**, labeled as **PoBMC**) from 16 PDAC patients showed CTC proliferation (mean ± SE) increased in *in vivo* and *ex vivo* formed clusters compared to FACS-isolated CTC cultured alone. Cultures containing FACS-isolated portal blood circulating tumor cells (labeled as **CTC**) and FACS-isolated Myeloid-derived Fibroblasts (labeled as **M-FB**) in defined co-cultures had comparable proliferation compared to PortalBMC cultured without separation

and increased proliferation than that seen in FACS-isolated CTC cultured alone. Error bars indicate standard error from the mean and ns indicates no significant difference found between the compared culture type means (bracketed). (**F**) From samples of 5 PDAC patients, FACS isolated, CFSE labeled CTC co-cultured with FACS isolated MitoTracker Red labeled M-FB isolated directly from PortalBMC were added alone or together in transwell cultures, migrated through transwells within 24hr at a 3.8-fold higher than CTC cultured separately. (**G**) Live cell micrograph depicted is representative of FACS-isolated *in vivo* viable clusters containing both CTC+ and MFB+ biomarkers collected directly from PDAC patient PortalBMC without *ex vivo* culture. Image was taken through a chamberwell slide within 12hr of blood draw and FACS isolation, using 20x magnification. (**H**) To visualize the interaction of FACS isolated single CTC with FACS isolated single M-FB population in culture, both isolated cell populations from 12 PDAC patient PortalBMC were labeled with vital dyes (CTC with CSFE, M-FB with MitoTracker Red) and cultured together at 37˚C/ 5%$CO_2$. When FACS-isolated CTC (green, CSFE) and FACS-isolated M-FB (red, MitoTracker Red) were analyzed by fluorescence microscopy, mixed CTC/M-FB clusters were detected forming within 24 hours. Dye overlap (white) seen in 40x magnification 2-D fluorescence microscopy may be indicative of cell overlap in the aggregates or dye transfer between cells, but could not be confirmed as cytoplasmic transfer or co-localization due to the transient nature of dye visualization in viable cells and the inability to retain the staining by fixation for 3-D confocal re-construction.

(**Fig 1A** & [11]), only CTC (**Fig 1B**) and M-FB (**Fig 1C**), were necessary for their formation (**Fig 1D**).

## PDAC CTC/M-FB clusters support portal CTC proliferation and migration

When CTC and M-FB were first FACS isolated and then co-cultured *ex vivo*, they supported CTC proliferation (**Fig 1E**) and promoted CTC migration (**Fig 1F**) compared to CTC cultured alone. Pre-operative (neo adjuvant) chemotherapy and/or radiation therapy did not significantly influence these interactions.

## PDAC CTC & M-FB clusters form in vivo in portal blood and within 24hr in ex vivo co-culture

By using a 100 micron nozzle in FACS isolation prior to culture, we found clusters containing portal CTC and M-FB pre-existing *in vivo* (**Fig 1G**). Vital dye incorporation in CTC and M-FB isolated by FACS prior to culture was used to track early cluster interactions of CTC (green, CSFE) and M-FB (red, MitoTracker Red). In defined mixed cultures of these FACS-isolated, fluorescently-labeled CTC and M-FB, CTC/M-FB clusters could form within 24 hours in culture (**Fig 1H**). Cytoplasmic exchange between CTC and M-FB as previously reported between fluorescent protein-transformed tumor cells and macrophages [10] may have been possible in our defined cultures but could not be definitively identified in 2-D fluorescent microscopy of these vital dye stained cells. The visualization of vital dye staining in CTC and M-FB clustering was transitory in viable cells and not amenable to fixation for preservation; therefore, cytoplasmic dye exchange between the participating cell types was unable to be confirmed by 3D reconstruction microscopy. Nevertheless, these findings support previous reports that direct cell-to-cell contact between CTC and myeloid cells can support CTC proliferation and migration [7–12].

## PDAC patient plasma contains high levels of myeloid chemokine, inflammatory, and growth factors

To investigate how CTC could be attracting myeloid cells and affecting myeloid differentiation to M-FB, plasma samples from intraoperative collection of portal and matched peripheral blood of PDAC patients were analyzed for cytokine and chemokine factors that may attract and support myeloid cell differentiation (**Fig 2**). GM-CSF levels in PDAC patient portal blood plasma were highly elevated compared to normal peripheral blood (p = 0.0305) while M-CSF

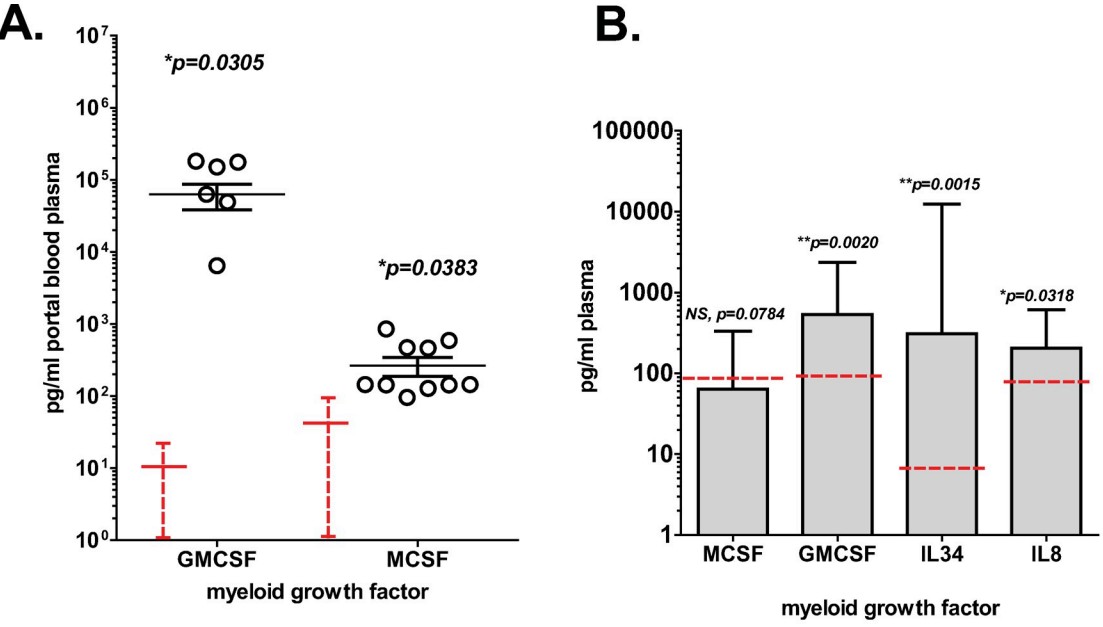

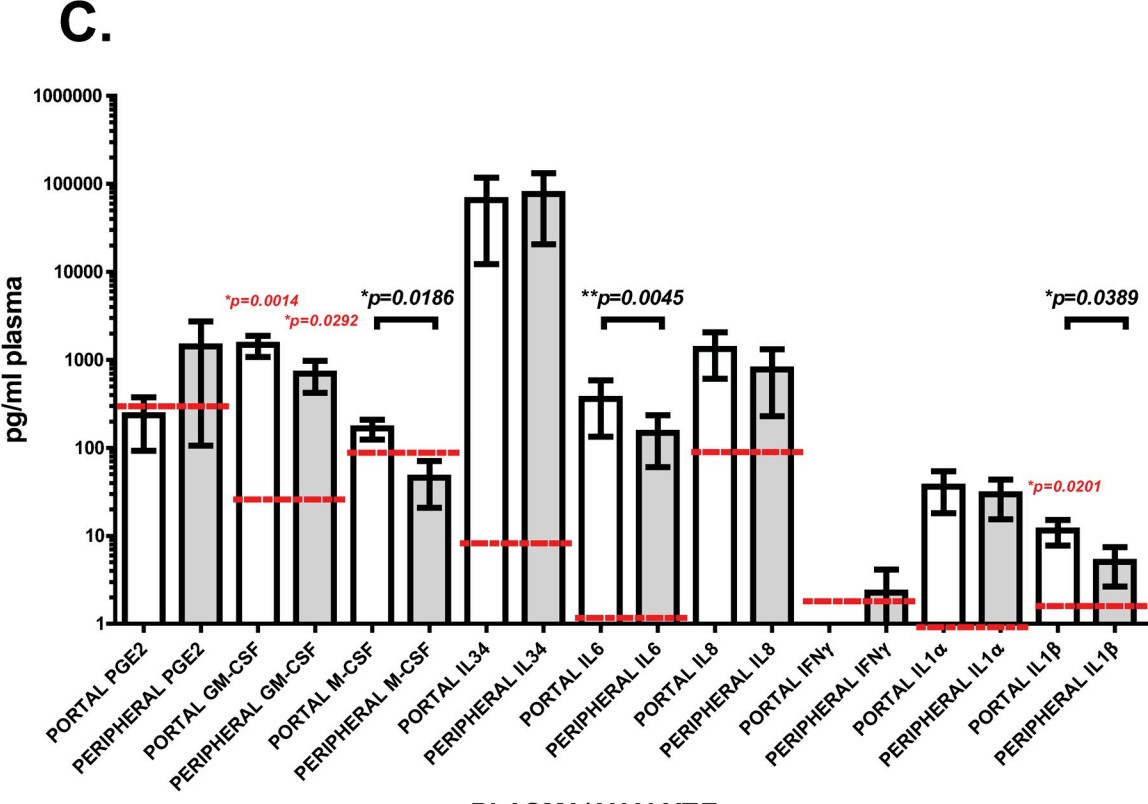

**Fig 2. Increased levels of inflammatory and myeloid cell growth factors in PDAC portal blood plasma.** (**A**) Portal blood plasma from 11 patients collected immediately after PDAC tumor resection were analyzed in duplicate using Legendplex cytokine multiplex flow cytometric analysis. By this analysis, GM-CSF levels in PDAC patient portal blood plasma were found highly elevated compared to normal peripheral blood levels (depicted by red dotted line mean + SD, p = 0.0305). M-CSF levels were moderately elevated over normal levels (red line and dotted line depicts control mean + SD, p = 0.0383). Sample values are depicted as open circles, lines indicate mean of sample values with error bars showing standard error from the mean. (**B**) Portal blood plasma from 24 PDAC patients were analyzed for

M-CSF, GM-CSF, and IL-34 by ELISA, indicating that the median levels of IL-34, IL-8, and GM-CSF in PDAC plasma (p = 0.0015, p = 0.0348, and p = 0.0020, respectively), were highly elevated in both PDAC portal (**B**) and peripheral (**C**) plasma compared to established levels seen in healthy peripheral blood plasma reported in the literature (healthy peripheral blood median depicted by red dotted lines). (**C**) Comparison of factor profiles in 19 matched PDAC patient portal (white bars) and peripheral (grey bars) plasma with adequate volume extended analysis showed significantly higher levels in portal blood of M-CSF (p = 0.0186), IL-6 (p = 0.0045), and IL-1β (p = 0.0389), while levels of GM-CSF, IL-34, IL-8, IL-1α, and PGE2 were elevated in both portal and peripheral plasma. Interferon gamma (IFNγ) was detectable only in peripheral blood plasma. Red lines indicate literature values for concentrations in non-malignant human plasma. The p values shown in black indicate where significant differences were found between PDAC portal and peripheral blood plasma samples of this study. The p values in red indicate where differences between portal or peripheral plasma samples were found significantly different from literature cited non-malignant control values. Error bars on all graphs indicate standard error from mean. * indicates p<0.05, ** p<0.01, and NS, no significant difference found (p>0.05).

levels were moderately elevated over normal (p = 0.0383, **Fig 2A**). By ELISA, the median/mean concentration levels of IL-34, IL-8, and GM-CSF in PDAC plasma (mean, p = 0.0015, median, p = 0.0348, and mean, p = 0.0020, respectively, **Fig 2B**) were elevated in both PDAC portal and peripheral plasma compared to established levels seen in healthy peripheral blood plasma reported in the literature (means ±SD depicted in red in **Fig 2A** and medians or means depicted by red dotted lines in **Fig 2B and 2C**). In comparisons of factor profiles in matched PDAC patient portal and peripheral plasmas showed significantly higher levels in portal blood of M-CSF (p = 0.0186), IL-6 (p = 0.0045), and IL-1β (p = 0.0389) than matched peripheral blood, while levels of GM-CSF, IL-34, IL-8, IL1α, and PGE2 were elevated equally in both portal and peripheral PDAC patient plasma. IFNγ was detectable in PDAC peripheral blood plasma but not in portal blood plasma (**Fig 2C**). Neo adjuvant pre-operative chemotherapy and/or radiation therapy did not significantly alter these comparisons. These findings suggest that the chemokine IL-8 and soluble myeloid growth and inflammatory factors could be promoting CTC interactions with myeloid cells brought within their sphere of influence.

## Phenotypic analysis of myeloid signaling receptors and factors on CTC cluster formation, proliferation, and apoptosis

Since both M-CSF and IL-34 signal through the CSF1R receptor, inhibitors against CSF1R could block these factors ability to promote cell growth and effects on myeloid differentiation [36]. Similar blockade of CSF2R could block GM-CSF signaling which promotes CSF1R expression, inflammation, and macrophage/dendritic cell differentiation [13, 26]. Using commercially available mouse and rabbit antibodies, we tested the effects of CSF1R and CSF2R inhibition on portal blood CTC cluster formation interactions (**Fig 3**). Addition of GM-CSF, M-CSF, or antibodies against the GM-CSF receptor (anti-CSF2R) to unsorted PortalBMC cultures did not significantly change mean CTC proliferation compared to medium only control. In contrast, anti-CSF1R treatment gave a 5.2-fold decrease in CTC proliferation in PDAC CTC PortalBMC cultures (p<0.0001, **Fig 3A**). PDAC PortalBMC formed clusters in medium alone (**Fig 3B**) and was enhanced in medium supplemented with GM-CSF (**Fig 3C**) or M-CSF (**Fig 3D**). In contrast, cluster formation is disrupted by the addition of anti-CSF1R antibodies (**Fig 3E**).

## Blockade of CSF1R, IL-34, and IL-8 signaling affects CTC cluster formation, proliferation, and apoptosis

To further investigate the role of CSF1R signaling in CTC-M-FB interactions, we used Bristol Myers Squibb humanized anti-CSF1R antibodies (BMS-986227, Cabiralizumab) to block the CSF1R and commercially available anti-IL34 antibodies (Abcam ab101443) to block IL-34 specific signaling. To block IL-8 chemokine signaling, we used the Bristol Myers Squibb humanized neutralizing anti-IL-8 antibody BMS-986253 with and without anti-CSF1R or anti-IL-34

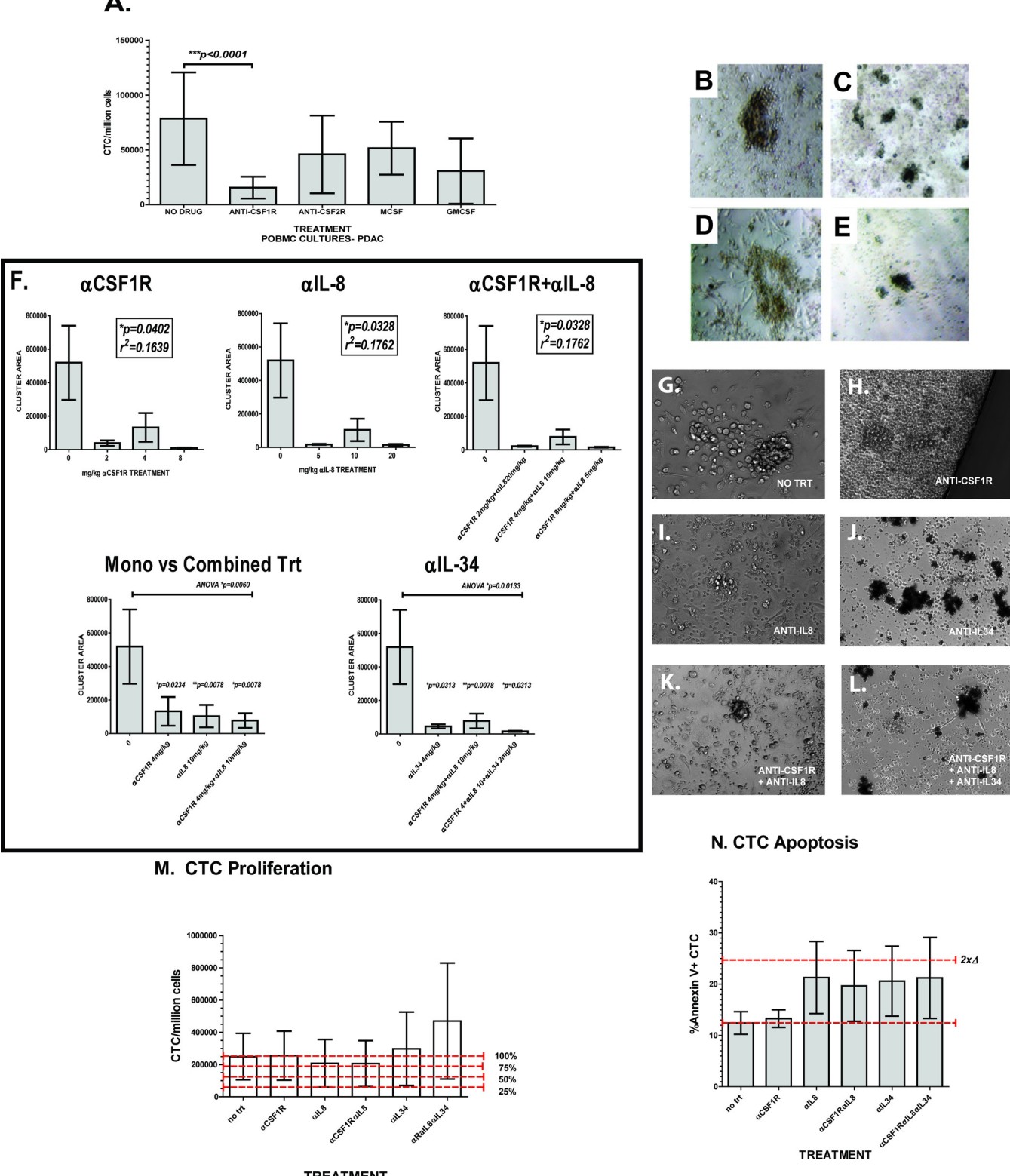

**Fig 3. CSF1R and IL-8 mediated signaling affect PDAC portal blood CTC proliferation, cluster formation, and apoptosis.** Sixteen PDAC patient PortalBMC unsorted samples were grown in *ex vivo* culture for 7 days with or without 1µg/million PortalBMC rabbit anti-CSF1R antibodies, 1µg/million PortalBMC mouse anti-CSF2R antibodies, 0.5 µg/ml M-CSF or 5nM GM-CSF. (**A**) Addition of GM-CSF, M-CSF, or antibodies against the GM-CSF receptor (**ANTI-CSF2R**) to unsorted PortalBMC cultures did not significantly change mean CTC proliferation compared to untreated cultures (**NO DRUG**). Anti-

CSF1R treatment gave a 5.2-fold decrease in CTC proliferation in unsorted PortalBMC cultures (p<0.0001). PDAC PortalBMC formed clusters in medium alone (**B**) and was enhanced in medium supplemented with GM-CSF (**C**) or M-CSF (**D**), but cluster formation was disrupted by the addition of rabbit anti-CSF1R antibodies (**E**). Images representative of clusters seen in PDAC 7 day cultures imaged at 20x magnification on a Zeiss Axiovert DIC microscope. PortalBMC cultures from 6 PDAC patients were treated with Bristol Myers Squibb humanized anti-CSF1R (BMS-986227, Cabiralizumab), Bristol Myers Squibb humanized anti-IL-8 (BMS-986253), and/or commercially available mouse anti-IL34 (Abcam ab101443) (**F-L**). CTC cluster formation in unsorted PortalBMC *ex vivo* cultures was disrupted by all 3 antibodies compared to (**F&G**) untreated PDAC *ex vivo* PortalBMC cultures. (**F**) Area of clusters was significantly decreased with all treatments (Depicted are 4mg/kg humanized anti-CSF1R (**H**), 10mg/kg anti-IL-8 (**I**), 4mg/kg anti-IL-34 (**J**), combination of 4mg/kg humanized anti-CSF1R and 10mg/kg anti-IL8 (**K**), and all 3 antibodies combined (**L**, 4mg/kg humanized anti-CSF1R, 10mg/kg humanized IL-8, 4mg/kg mouse anti-IL-34). Correlations were seen between decreased CTC cluster size with concentration of treatment with humanized antibodies (**F**, anti-CSF1R and anti-IL-8, alone or in combination), but the relationships were not linear. Images (**G-L**) and Image J area analysis (**F**) depicted are representative of PortalBMC clusters seen in 7 day cultures imaged at 20x magnification on a Zeiss Axiovert DIC microscope. In comparison with PDAC PortalBMC cultures treated with rabbit anti-CSF1R antibodies (**A**), CTC proliferation was only modestly decreased (**M,** n = 6) in cultures treated with humanized 10mg/kg anti-IL-8 (BMS-986253), alone or in combination with humanized 4mg/kg anti-CSF1R. However, CTC apoptosis (**N,** n = 3) was markedly increased in cultures treated with humanized 10mg/kg anti-IL-8, alone or in combination with humanized 4mg/kg anti-CSF1R (BMS-986227, Cabiralizumab). Treatment with 4mg/kg anti-IL-34 alone or in combination with 4mg/kg humanized anti-CSF1R and 10mg/kg humanized anti-IL-8 enhanced both CTC proliferation and apoptosis (**M&N**). For graphs in panels **A, F, M,** and **N,** bars depict mean of values and error bars indicate standard error from the mean.

blockade. All 3 antibody treatments alone or in combination disrupted CTC cluster formation compared to untreated PDAC *ex vivo* PortalBMC cultures (**Fig 3F–3L**). CTC proliferation was modestly decreased (**Fig 3M**) and CTC apoptosis markedly increased (**Fig 3N**) in cultures treated with humanized anti-IL-8 (BMS-986253), alone or in combination with anti-CSF1R (BMS-986227, Cabralizumab). In contrast, treatment with anti-IL-34 alone or in combination with anti-CSF1R and anti-IL-8 variably enhanced both CTC proliferation and apoptosis (**Fig 3M and 3N**).

## Treated PDAC PortalBMC culture-generated conditioned media affected myeloid differentiation of U937 myeloid precursor cell line

We examined the phenotype diversity of myeloid cells within the PortalBMC *ex vivo* cultures of 5 PDAC patients and found M-FB CD14+CD105+ constituted the majority of myeloid cells detected at 7 days in culture. M-FB constituted 29.7%± 43.6SD of myeloid cells compared with 4.7%± 8.6SD for CD14+DR+ macrophage, 1.5%± 1.6SD CD11c+DR+CD14- dendritic cells, and 0.70%± 0.50SD CD14+CD33+DR- MDSC. Since MDSC can be induced to form fibroblast-like morphology with excess M-CSF and the level of CSF1R signaling can both enhance and inhibit differentiation pathways in myeloid development [18, 36], we tested the potential for *ex vivo* culture conditioned media of anti-CSF1R, anti-IL-8, and anti-IL34 treated PortalBMC to affect myeloid differentiation.

We collected media from 7-day PDAC patient PortalBMC cultures that had been treated with anti-CSF1R (Cabiralizumab), anti-IL-8 (BMS-986253) and/or anti-IL-34, and applied these conditioned media to cultures of the human pre-myeloid cell line U937 and assayed the cultures after 7 days for biomarkers of myeloid differentiation to macrophage, dendritic cells, MDSC, and M-FB. Conditioned media from PortalBMC cultures treated to inhibit CSF1R signaling alone or in combination with IL-8 blockade still allowed U937 cells to undergo myeloid differentiation to MDSC and MF-B, and to antigen presenting cell (APC) phenotypes MHCII DR+ macrophages and dendritic cells. In contrast, conditioned media from PDAC PortalBMC cultures treated with anti-IL-34 significantly blocked U937 myeloid differentiation in all but the M-FB differentiation phenotype. Conditioned media from PortalBMC cultures treated with anti-IL-34 significantly decreased U937 M-FB phenotype differentiation compared to cultures treated with conditioned media of PortalBMC cultures grown without any of the antibody inhibitors ('**zero**', p = 0.0131, **Fig 4**). Anti-IL-34 treated media suppression of U937 M-FB differentiation was partially restored in conditioned media from PortalBMC treated with combined anti-IL-34, anti-CSF1R and anti-IL-8, allowing for M-FB differentiation but

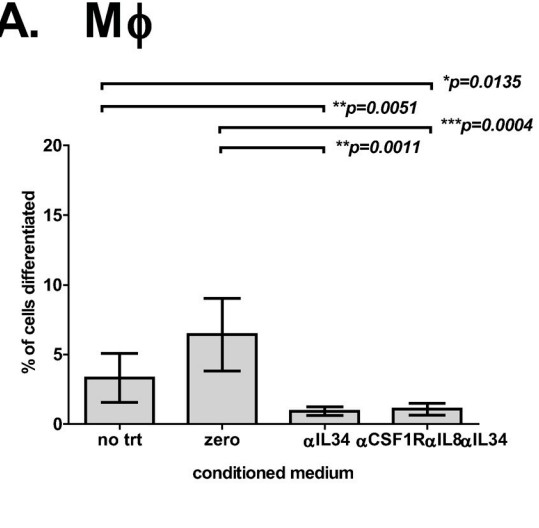

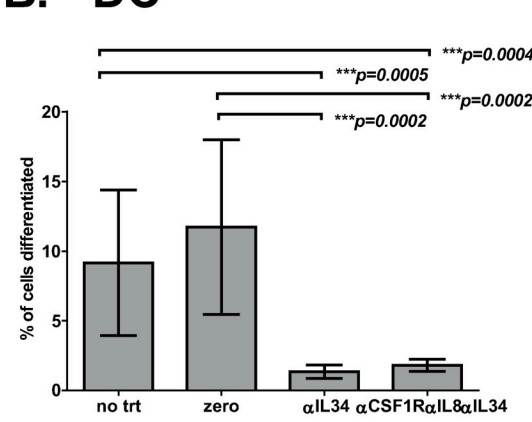

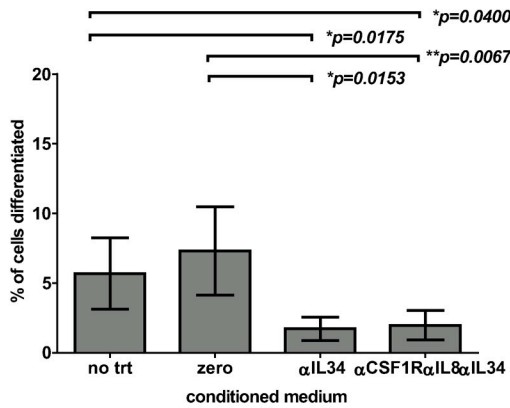

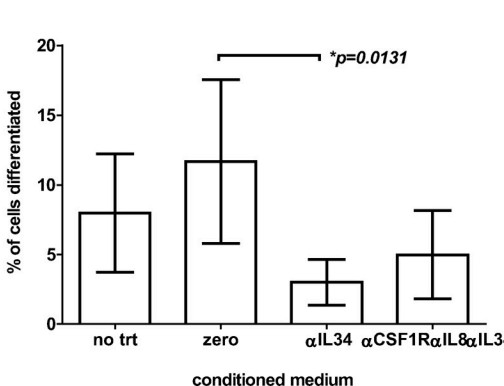

**Fig 4. Myeloid precursor cell line U937 differentiation affected by PDAC PortalBMC conditioned media.** Conditioned media collected from 8 PDAC patients' 7-day PortalBMC cultures were used to stimulate myeloid differentiation in cultures of the human pre-myeloid cell line U937 (ATCC). The myeloid differentiation seen in U937 cells cultured for up to 7 days supplemented with conditioned media from PortalBMC cultures treated with 4mg/kg humanized anti-CSF1R (Cabiralizumab), 10mg/kg humanized anti-IL-8 (BMS-986253) or 4mg/kg anti-IL-34 alone or in combination was compared to that seen in U937 cultures treated with media from untreated PortalBMC cultures (**zero**) and U937 cells cultured without conditioned media (**NO TRT**). Myeloid differentiation was determined by flow cytometry biomarker identification using the myeloid subpopulation profiles: (**A**) CD14+DR+ macrophages (**Mφ**), (**B**) CD11c+DR+CD14- myeloid dendritic cells (**DC**), (**C**) CD14+CD33+DR-CD11c- myeloid-derived suppressor cells (**MDSC**), and (**D**) CD14+CD105+ Myeloid-derived fibroblasts (**M-FB**). Conditioned media from untreated cultures (**zero**) and humanized anti-CSF1R, humanized anti-IL-8 or anti-CSF1R+anti-IL-8 treated cultures did not statistically decrease the myeloid differentiation of U937 cells to Macrophage, Dendritic Cell, MDSC, and M-FB phenotypes. (data not shown) Conditioned media from PortalBMC cultures treated with anti-IL-34 alone or in combination with humanized anti-CSF1R and humanized anti-IL-8 significantly blocked myeloid differentiation in all (**A-C**) but (**D**) M-FB phenotypes. Conditioned media from PDAC PortalBMC cultures treated with anti-IL-34 inhibition alone significantly decreased the M-FB differentiation phenotype by compared to conditioned media from untreated (**zero**) cultures (**D**). PortalBMC culture anti-IL-34 conditioned media suppression of U937 M-FB differentiation was partially restored in U937 cultures supplemented with conditioned media from PortalBMC cultures treated with anti-IL-34 in combination with anti-CSF1R and anti-IL-8 (**D**). The statistically significant different p values for pairwise analyses of % phenotypic cells between treated and either untreated or 'zero' media-treated U937 cultures are listed above each graph with bars indicating the compared culture sets. Bars depict mean of values and error bars indicate standard error from the mean.

not myeloid APC or MDSC differentiation. These data suggest that IL-34 is a primary soluble factor produced in PortalBMC cultures that can affect myeloid differentiation. Considering IL-34 and M-CSF compete for binding of the CSF1R receptor on myeloid cells, with IL-34 having a higher CSF1R binding affinity [26], the predominance of M-FB phenotypic cells in PDAC PortalBMC may reflect this unequal competition at the CSF1R receptor and/or the ability of IL-34 to signal through other unaffected receptors on myeloid cells and CTC, including PTPζ and sydecan-1 that can also influence migration and CTC proliferation [37].

## CSF1R and CXCR2 receptor expressed on PDAC PortalBMC CTC and M-FB

To establish which cells in the CTC/M-FB co-cultures were able to receive IL-8, IL-34, and M-CSF signals, the level of protein expression of CXCR2 and CSF1R receptors was analyzed by flow cytometry on both CTC and M-FB cell populations FACS isolated out of uncultured PortalBMC and after 7-day in *ex vivo* culture. The expression of CXCR2 and CSF1R proteins was detected on the surfaces of PDAC portal blood CTC and M-FB in flow cytometry prior to and after culture. When compared by mean fluorescence of the labeled receptors, none of the treatments significantly altered either CXCR2 or CSF1R expression on CTC and M-FB.

## M-CSF, IL-8, and IL-34 levels are elevated in PDAC PortalBMC culture supernatants

Conditioned medium from PDAC PortalBMC cultures were analyzed for M-CSF, IL-8, IL-34, and GM-CSF (**Fig 5**). Media from untreated PortalBMC cultures contained elevated levels of IL-34 and GM-CSF comparable to the levels seen in portal blood plasma (**Figs 2A and 5C, 5D**), while M-CSF and IL-8 levels were higher in the culture supernatants than detected in plasma (**Figs 2B and 5A, 5B**). When compared with the levels detected in untreated cultures, cultures treated with anti-CSF1R (BMS-986227, Cabiralizumab) suppressed production of M-CSF (**Fig 5A**). Treatment with anti-IL-8 (BMS-986253), suppressed IL-8 and M-CSF production (**Fig 5A and 5B**). Combined anti-IL-8/anti-CSF1R or anti-IL-8/anti-CSF1R/anti-IL-34 treatments enhanced these effects. Cultures treated with anti-IL34 antibody alone or in combination suppressed M-CSF and IL-34 production (**Fig 5A and 5C**). No treatment altered the GM-CSF levels seen in these cultures (**Fig 5D**).

## RNA expression is elevated for IL-8, M-CSF, and IL-34 in PDAC PortalBMC CTC and CTC clusters

To test whether CTC or M-FB could be sources of the elevated IL-8, IL-34, M-CSF, and GM-CSF found in the portal blood plasma and PortalBMC culture supernatants, CTC, M-FB, and CTC & M-FB containing clusters found formed *in vivo* were FACS isolated from PortalBMC without culture and directly extracted for analysis of their RNA expression (**Fig 6**). *CSF1* RNA expression was elevated in isolated CTC and *CSF2* (GM-CSF) RNA expression was elevated *in vivo* formed CTC+/M-FB+ clusters (**Fig 6A**). High variability was seen in isolated single cell clone (SCS) CTC *CSF1* RNA expression in RNA SEQ and in real-time RT-PCR suggestive of clonal variation within individual patient portal blood tumor cell populations (data not shown). *CSF2* expression was higher in *in vivo* clusters compared to CTC alone (p = 0.0322, **Fig 6A**). Though found elevated in both, *IL34* RNA expression was higher in isolated CTC than in *in vivo* clusters (p = 0.0186, **Fig 6A**). The M-CSF/IL-34 receptor *CSF1R* mRNA expression was detectable in CTC, *in vivo* clusters, and M-FB, with the expression

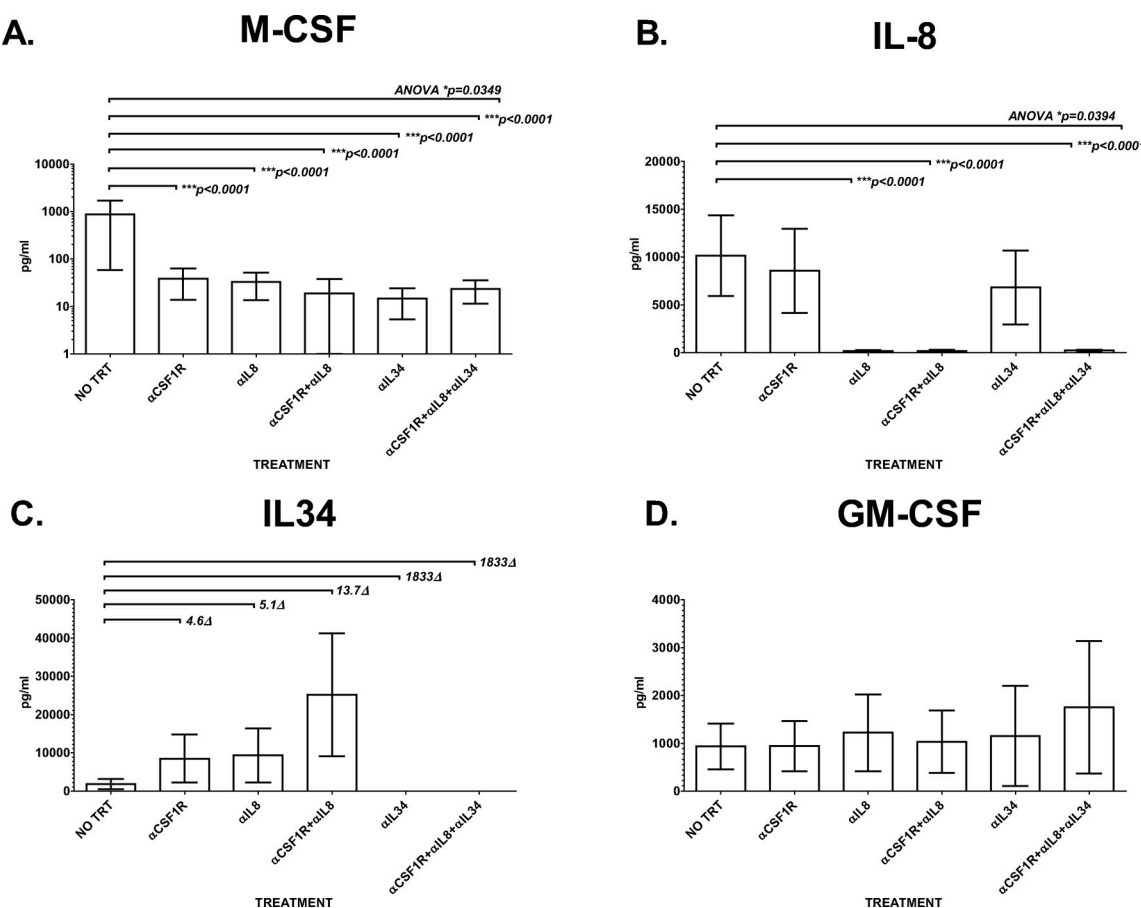

**Fig 5. High M-CSF, IL-8, IL-34, and GM-CSF levels were found in conditioned media from 7-day PDAC PortalBMC cultures.**
Culture media supernatants from 7 day cultures of 8 PDAC patients *ex vivo* PortalBMC were analyzed in duplicate by ELISA for (**A**) M-CSF, (**B**) IL-8, (**C**) IL-34, and (**D**) GM-CSF. Untreated cultures had elevated levels of M-CSF and IL-8 and comparable levels of IL-34 and GM-CSF compared to PDAC portal blood plasma and elevated levels of all 4 analytes compared to healthy human plasma analyzed by ELISA (Fig 2A). (**A**) Treatment with 4mg/kg humanized anti-CSF1R (BMS-986227, Cabiralizumab) suppressed production of M-CSF (CSF1). Treatment with 10mg/kg humanized anti-IL-8 (BMS-986253), suppressed CSF1(**A**) and IL-8 (**B**) production. Combined 10mg/kg anti-IL-8 /4mg/kg anti-CSF1R or 10mg/kg anti-IL-8/4mg/kg anti-CSF1R/4mg/kg anti-IL-34 treatments enhanced these effects. (**C**) Cultures treated with anti-IL34 antibodies alone or in combination suppressed M-CSF and IL-34 production. (**D**)No treatment significantly altered the GM-CSF levels seen in these cultures. The p values for significantly different pairwise and ANOVA comparisons are listed by the brackets indicating the test groups compared to untreated cultures (**NO TRT**). Bars depict mean of values and error bars indicate standard error from the mean.

highest but most variable in M-FB (**Fig 6B**). *CSF2R* RNA expression was not detected in these cell and cluster samples.

Further analysis of RNA expression of CTC and M-FB cells that were FACS isolated from untreated PortalBMC after culture suggest CTC but not M-FB express elevated RNA for *CXCL8 (IL8)* and *IL-34* (**Fig 7**). Both CTC and M-FB had elevated RNA expression of *CSF1R* and *CSF1* after culture (**Fig 7A and 7D**). Treatment of PortalBMC cultures with humanized anti-CSF1R (BMS-986227, Cabiralizumab) significantly suppressed *CSF1R* and *CXCL8* RNA expression in CTC (**Fig 7A and 7C**). *CSF2* RNA expression appeared activated in CTC in untreated cultures but was suppressed by humanized anti-IL-8 (BMS-986253) or anti-IL-34 treatments (**Fig 7B**). Treatment of the cultures with humanized anti-IL-8 in combination with humanized anti-CSF1R suppressed *CXCL8* and *CXCR2* RNA expression in CTC (**Fig 7C**). Treatment with anti-IL-34 enhanced *CSF1R*, *CSF1*, and *IL34* expression in CTC, while

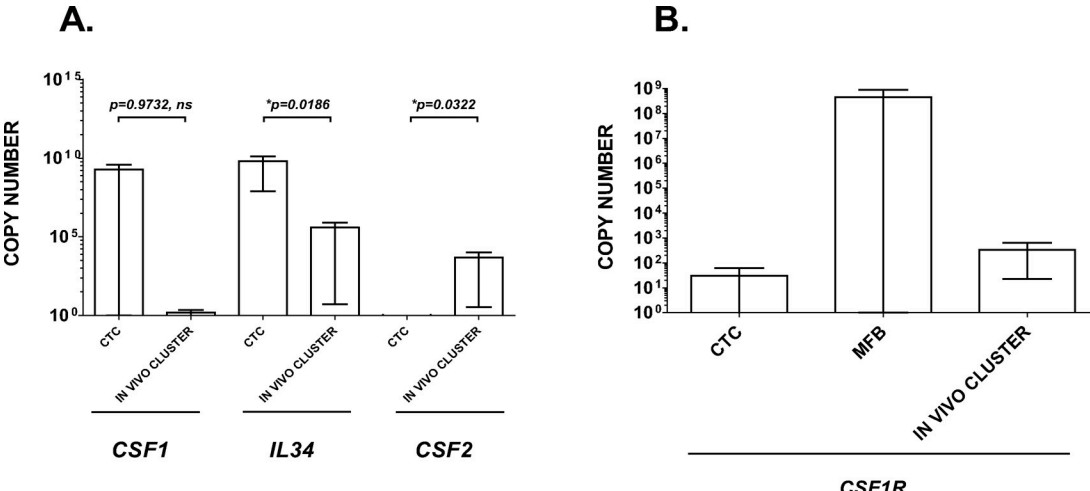

**Fig 6. RNA expression of myeloid growth factor signaling ligands and their receptors in uncultured PDAC CTC and M-FB isolated cells and *in vivo* formed clusters.** Uncultured portal blood CTC, M-FB, and *in vivo* formed CTC+/M-FB+ clusters from PortalBMC of 6 PDAC patients were FACS isolated and collected into RNALater, and then extracted and analyzed for *CSF2*, *CSF1*, *IL34*, *CSF1R*, and *CSF2R* RNA expression by real-time RT-PCR. (**A**) *CSF1* (M-CSF) RNA expression was elevated in isolated CTC and *CSF2* (GM-CSF) RNA expression was elevated *in vivo* CTC+M-FB clusters. High variability was seen in isolated CTC *CSF1* RNA expression, complicating interpretation of statistical analyses. *CSF2* expression was higher in *in vivo* clusters compared to CTC alone (p = 0.0322). Though found elevated in both, *IL34* RNA expression was higher in isolated CTC than in *in vivo* clusters (p = 0.0186). (**B**) M-CSF/IL-34 receptor *CSF1R* mRNA expression was detectable in CTC, *in vivo* clusters, and M-FB, with the expression highest but most variable in M-FB. *CSF2R* RNA expression was not detected in this sample set. Due to limitations in uncultured cell isolate numbers, RNA for *CXCL8 (IL8)* and *CXCR2* were not analyzed. Bars depict mean of values and error bars indicate standard error from the mean.

suppressing expression of *CXCL8* RNA (**Fig 7A and 7C**). Combining all 3 antibody treatments (anti-CSF1R/anti-IL-8/anti-IL-34) together was needed to suppress *IL-34* RNA expression in CTC (**Fig 7A**).

In contrast to CTC, RNA expression in all 3 signaling pathways, *CSF1/IL34/CSF1R*, *CSF2/CSF2R*, and *CXCL8/CXCR2*, appeared to be unaffected or enhanced in M-FB by anti-CSF1, anti-IL34, or anti-IL-8 alone or combined treatments, with the exception of significant decreases in *CSF1R* RNA expression with anti-CSF1R or anti-IL-8 as single treatments. Of note, M-FB RNA expression of *IL34* was not elevated and *CXCL8* RNA expression was not detected in untreated PortalBMC culture M-FB isolates. (**Fig 7D–7F**)

Although we cannot rule out that other cell types remaining in the PortalBMC cultures as possible sources of these factors, these data indicate that PDAC CTC can be a major contributor to the high concentrations of IL-34, M-CSF, GM-CSF, and IL-8 expression in the portal blood tumor microenvironment.

Taken together, these findings suggest that PDAC CTC can contribute to myeloid chemoattraction, differentiation, and growth factor signaling pathways, potentially enhancing the ability of myeloid cells to respond to these signals in the portal blood. Our findings suggest that combined anti-IL8 and anti-CSF1R treatments disrupts CTC chemoattraction of myeloid cells and diminishes signaling of M-CSF and IL-34 through CSF1R. However, CTC-produced IL-34 could still signal through other receptors to promote CTC proliferation and migration [37]. Blocking chemoattraction by IL-8 could blunt these effects by diverting myeloid cells away from direct cell-cell interactions with CTC and allowing them to respond to immune-mediated signaling such as GM-CSF and TNFα/IL-4 that promote myeloid APC development. We speculate that this could in turn could mitigate immune quiescence and tumor cell support mediated by MDSC and M-FB.

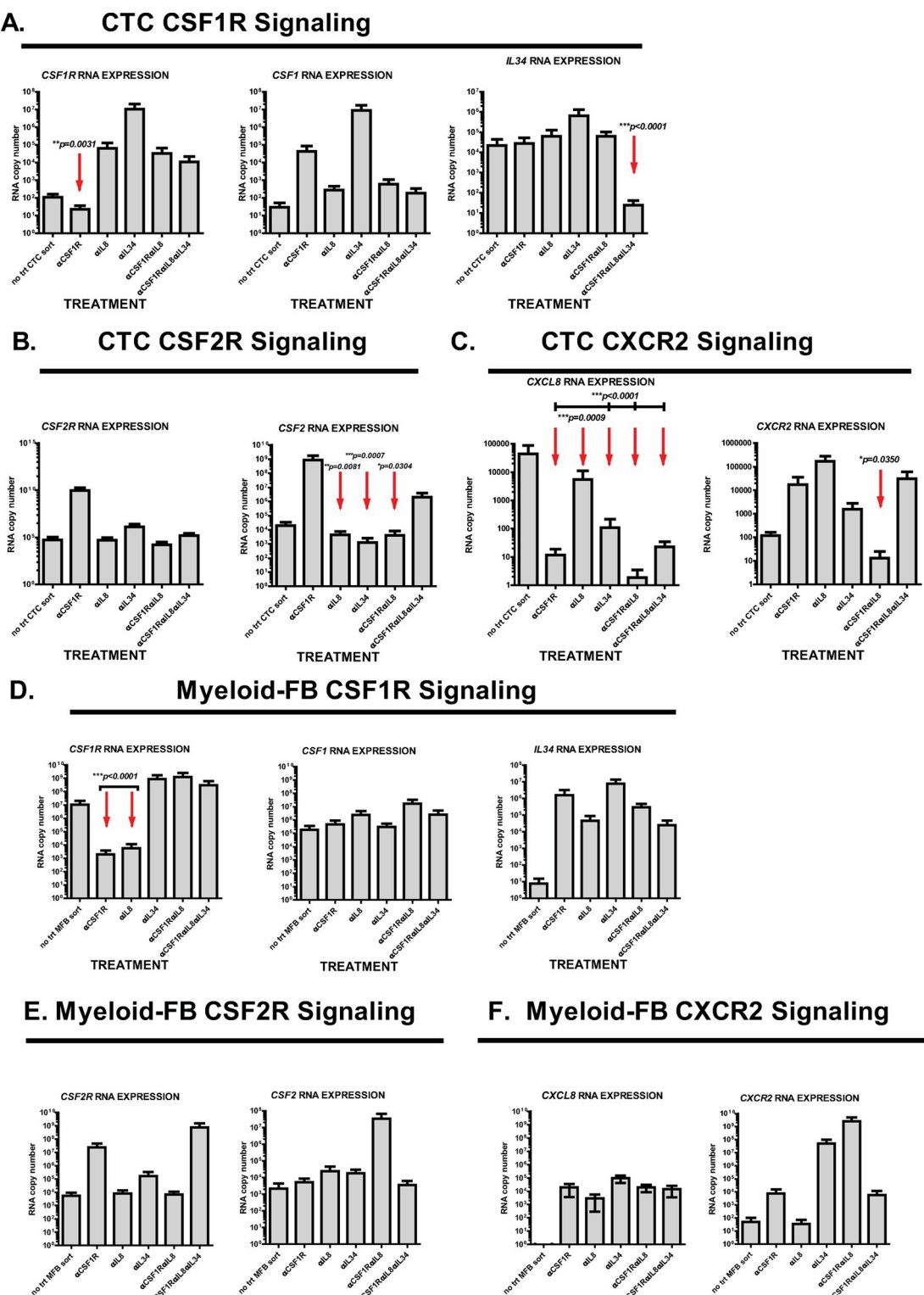

**Fig 7. RNA expression of myeloid growth factor signaling ligands and their receptors found in PDAC CTC and M-FB after PortalBMC *ex vivo* culture.** CTC and M-FB from PortalBMC of 6 PDAC patients were FACS isolated after 7 days in culture with or without (**no trt**) treatment with 4mg/kg humanized anti-CSF1R (BMS-986227, Cabiralizumab), 10mg/kg humanized anti-IL-8 (BMS-986253), 4mg/kg anti-IL-34 (Abcam). Isolated CTC and M-FB were extracted and analyzed for *CSF2*, *CSF1*, *IL34*, *IL8*, *CXCR*, *CSF1R*, and *CSF2R* by Real Time RT-PCR for RNA expression related to (**A&D**) CSF1R signaling (*CSF1R*, *CSF1*, *IL34*),

(**B&E**) CSF2R signaling (*CSF2R*, *CSF2*), and (**C&F**) CXCR2 signaling (*CXCL8*, *CXCR2*) in each cell type. (**A**) CTC RNA expression of *CSF1R*, *CSF1*, and *IL34* were elevated in untreated cultures and only anti-CSF1R treatment was able to suppress *CSF1R* expression (p = 0.0031) and only combined treatment of anti-CSF1R/anti-IL-8/anti-IL-34 was able to suppress *IL34* expression (p<0.0001). No significant decrease was seen in CTC *CSF1* RNA expression with any treatment, in contrast to effects seen at the *CSF1* gene product level (M-CSF/CSF1, **Fig 5**). (**B**) CTC *CSF2R* expression was not significantly altered by any treatment; however, the elevated expression of *CSF2* RNA in CTC from untreated cultures was suppressed by anti-IL-8, anti-IL34, and anti-IL8/anti-CSF1R treatments. (**C**) Substantial untreated PDAC PortalBMC culture expression of *CXCL8 (IL-8)* RNA was seen in CTC, which was suppressible by treatments with anti-CSF1R, anti-IL-8 and anti-IL-34, alone or in combination. In addition, CTC *CXCR2* RNA was only suppressed by combined anti-CSF1R/anti-IL-8 treatment. Red arrows indicate significant changes seen in statistical analysis. Bars depict mean of values and error bars indicate standard error from the mean. Arrows highlight where significant differences were seen between treated and untreated cultures. In contrast, RNA expression of *IL34* (**D**) was not elevated and *CXCL8 (IL-8*, **F***)* RNA expression was not detected in untreated culture M-FB FACS isolates, but all treatments appeared to activate their expression, though the increased levels were not found statistically significant. Indeed, M-FB RNA expression of all 3 signaling pathways, (**D**)*CSF1/IL34/CSF1R*, (**E**) *CSF2/CSF2R*, and (**F**) *CXCL8(IL8)/CXCR2*, appeared to be unaffected or enhanced by anti-CSF1, anti-IL34, or anti-IL-8 alone or as combined treatments, with the exception of significant decreases seen in *CSF1R* RNA expression with anti-CSF1R or anti-IL-8 as single treatments (**D**).

### PDAC Portal T cell anergy decreased and IFNγ levels increased with anti-CSF1R, anti-IL-8, and anti-IL34 treatments

Disruption of CTC influence on myeloid cell differentiation could result in a decrease in MDSC/M-FB phenotypes, while allowing more myeloid development toward MHCII-DR expressing APC, with the potential of replacing anergy inducing signals from MDSC with APC produced T cell activation signaling and promotion of anti-tumor activity. To investigate this possibility, we examined the anergic T cell phenotypes of T cells in PortalBMC cultures treated with anti-CSF1R, anti-IL-8 and anti-IL-34 (**Fig 8**). As previously noted in the portal blood [11], T cells found in the portal blood tumor microenvironment and in PortalBMC grown *ex vivo* are found in a highly anergic state (CD25+CTLA4+PDL1+PD1+CD45+CD3+, **Fig 8A**). However, treatment of PDAC PortalBMC cultures with anti-CSF1R, anti-IL-8, and/or anti-IL34 alone or in combination showed a 2-3-fold reduction of anergy among surviving PortalBMC T cells, though the number of samples we were able to analyze in these studies limited the statistical analysis of these results.

Portal blood plasma (**Fig 2C**) and conditioned media from untreated 7-day *ex vivo* PortalBMC cultures (**Fig 8B**) showed no detectable IFNγ, suggestive of the lack of activated T cells present. Treatment of cultures with humanized anti-CSF1R or humanized anti-IL-8 as single treatments did not significantly increase IFNγ production, but when used in combination, IFNγ was detected in the conditioned media (**Fig 8B**). Treatment with anti-IL-34 alone (p = 0.0307) or in combination with anti-CSF1R and anti-IL-8 (p = 0.0294), significantly enhanced IFNγ production, suggestive of increased T cell activation within the cultures (**Fig 8B**).

## Discussion

The portal venous system can act as a reservoir for PDAC CTC to adapt, survive, and grow, enhanced by CTC cluster formation. CTC cluster formation is dependent on the attraction and differentiation of myeloid cells into M-FB [7, 9, 11]. M-FB/CTC interactions facilitate CTC passage through the hostile immune environs of the portal blood, suppress tumor-directed immune responses, and support pro-metastatic CTC subpopulations migration out of the vessel and into tissues [10, 13, 15]. The signaling required for M-FB differentiation from MDSC precursors is poorly understood, including the mechanisms of how increased signaling through the myeloid differentiation/growth factor pathways allows for development of more fibroblast-like character and increased tumor supportive functions [10, 13].

We found abnormally high GM-CSF, IL-8, M-CSF, IL-6, IL-1β, and IL-34 concentrations in the portal blood of patients with PDAC. These factors can enhance myeloid cell migration

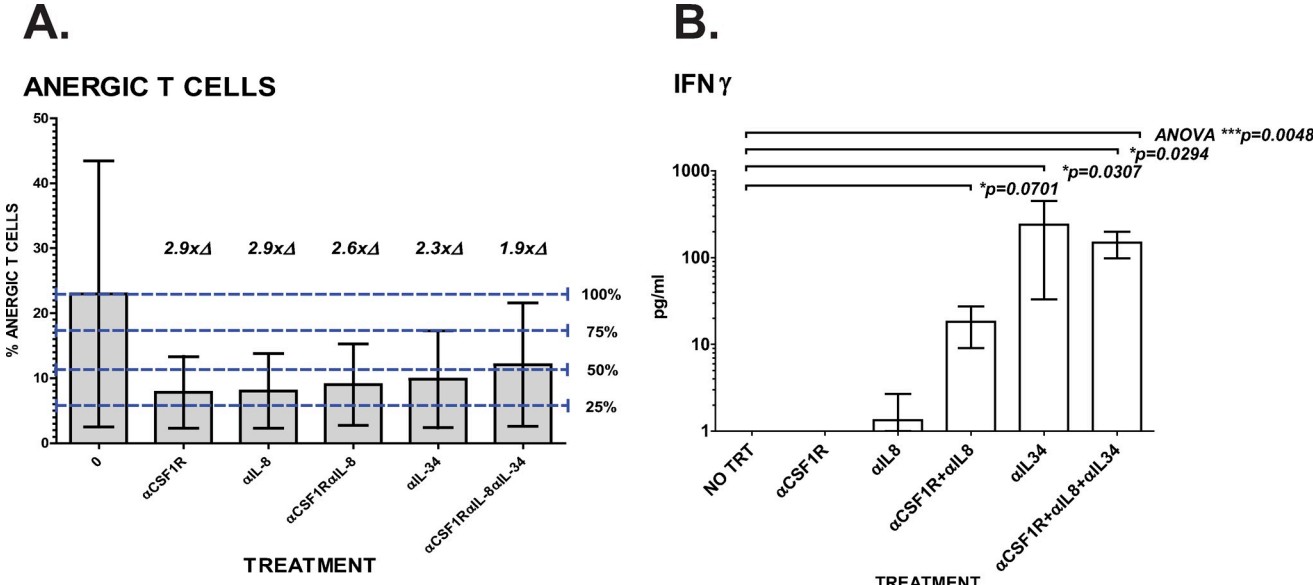

**Fig 8. Inhibition of CSF1R, IL-8, and IL-34 reduces T cell anergy in PortalBMC *ex vivo* cultures.** PortalBMC from 4 PDAC patients were cultured unsorted for 7 days at 37˚C/5%CO$_2$ in rich media with or without (**NO TRT**) supplementation with 4mg/kg humanized anti-CSF1R (BMS-986227, Cabiralizumab), 10mg/kg humanized anti-IL-8 (BMS-986253), and/or 4mg/kg anti-IL-34 (Abcam). After incubation, cells were collected and analyzed for identification of T cell subpopulations by flow cytometry and culture supernatant conditioned media were analyzed in duplicate by ELISA for comparison of IFNγ levels relative to untreated PortalBMC cultures. (**A**) Though the limited sample number precluded analysis of statistical significance, a strong trend for decreasing T cell anergy with antibody treatments was seen. Analysis of anergic T cells (CD25+CTLA4+PDL1+PD1+CD45+CD3+) in 7-day *ex vivo* PortalBMC cultures showed a 2-3-fold reduction of anergy among T cells in PortalBMC cultures treated with anti-CSF1R, anti-IL-8, and/or anti-IL34 alone or in combination. (**B**) Conditioned media from 6 PDAC patients' 7-day *ex vivo* PortalBMC cultures without treatment showed no detectable IFNγ, suggestive of a lack of activated, non-anergic T cells present. Portal plasma from these 4 PDAC patients as well as 4 others also had no detectable IFNγ (**Fig 2C**). In contrast, peripheral blood plasma samples from the same 8 PDAC patients had detectable IFNγ levels comparable to healthy controls (**Fig 2C**). Treatment of 4 PDAC PortalBMC cultures with humanized anti-CSF1R or humanized anti-IL-8 alone did not significantly increase IFNγ production, but treatment with the combination of 4mg/kg anti-CSF1R and 10mg/kg anti-IL-8 increased IFNγ concentration in the conditioned media (**B**), though statistical significance was not achieved. The treatment of cultures with 4mg/kg anti-IL34 alone or in combination with anti-CSF1R and anti-IL-8 showed significantly enhanced IFNγ levels (**B**, p = 0.0307 and p = 0.0294, respectively), suggestive of possible increased T cell activation within the cultures. Bars depict mean of values and error bars indicate standard error from the mean.

and promote differentiation signaling in myeloid precursor cells [11, 38–40]. These signal pathways appear to also support CTC clustering and survival in the portal circulation. Zhu and colleagues [39] found CSF1R expression in human PDAC tumor cells as well as intra-tumoral (TAM) macrophages, similar to what we see in PDAC CTC. Das et al [26, 41] contend that IL1β can promote tumorigenesis and cytotoxic T cell suppression in mouse models of PDAC. The abundance of IL1β along with the high myeloid influencing cytokine/chemokine milieu of PDAC portal blood could synergistically support CTC survival, both by acting directly on CTC and through influencing myeloid cell differentiation to M-FB which simultaneously support CTC cluster formation and block activation of anti-tumor immune responsive T cells (**Fig 9A**).

Li and Stanger [9] have suggested that in PDAC, clustered CTC may have a different role in tumor survival than individually shed CTC. Our data suggest that individual CTC clones could use M-CSF/IL-34 signaling to induce or promote M-FB differentiation to aid in cluster formation and myeloid support of extravasation to tissues. In contrast, CTC released from tumors as pre-formed clusters, where CTC and myeloid cells are already aggregated together, may use GM-CSF to alter the function of the myeloid cells toward a more stromal-like role that promotes inflammation and prevents cytotoxic T cell-induced apoptosis in the portal microenvironment. Recent reports on CTC detection in multiple cancer types have suggested

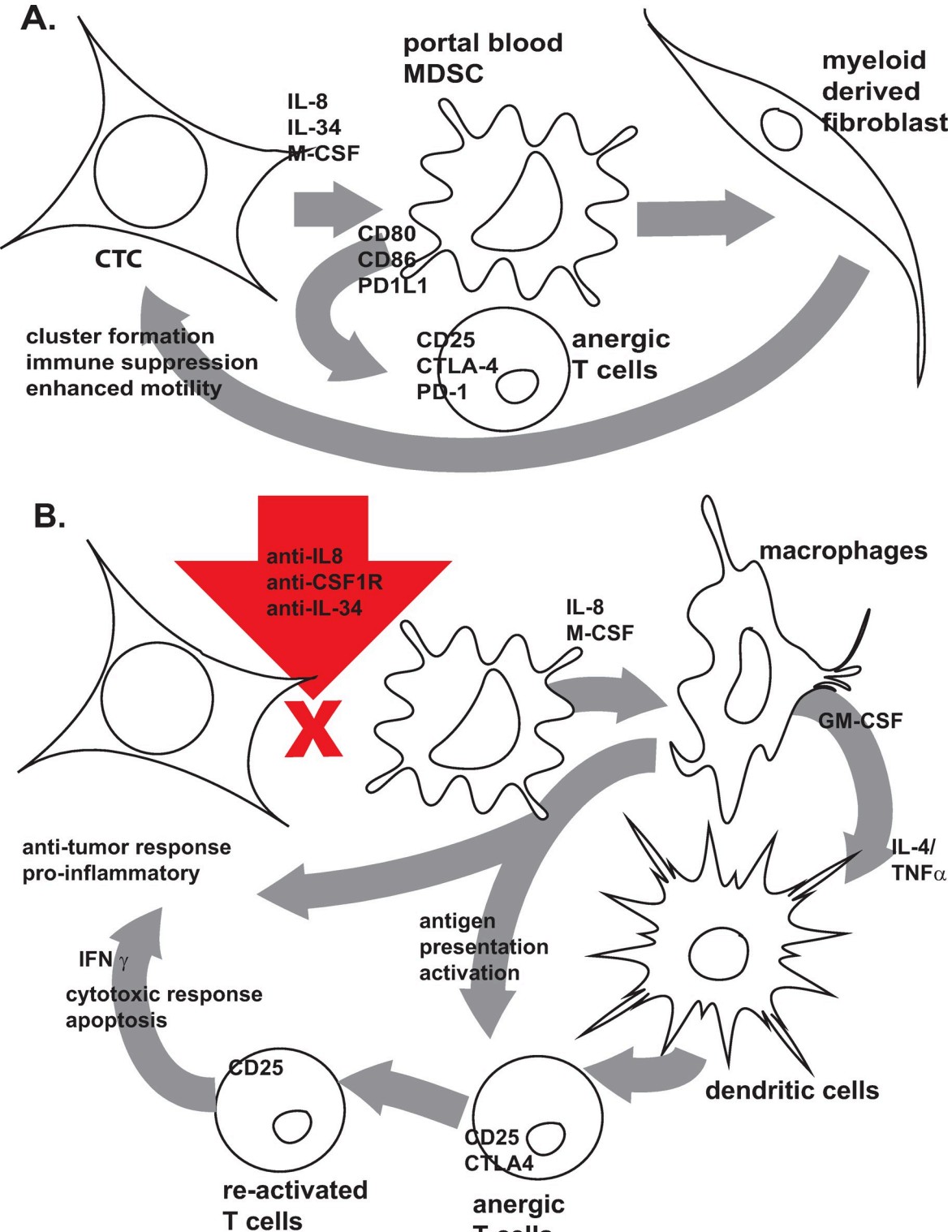

**Fig 9. Mechanism of action working model hypothesis: CTC influence on myeloid cell differentiation plays a pivotal role in CTC survival and T cell immune response. A)** PDAC CTC draw portal blood myeloid cells to them via IL-8 chemotaxis receptor CXCR2 signaling and then manipulate their differentiation to MDSC and further to M-FB through their production of high levels of IL-34 and M-CSF. These suppressor cell populations promote T cell anergy through PD-1 and CTLA4 as well as give CTC growth factors, metabolic support, motility enhancement, and immune suppressive protection. **B)** When IL-8-CXCR2 and IL-34/M-CSF-CSF1R signaling are

blocked, CTC are left without the support of MDSC and M-FB. This allows myeloid precursor cells and monocytes in the portal blood to differentiate away from CTC influence, allowing them to respond to their own and other immune cell signaling for development into activated antigen presenting cells (APC) including inflammatory activated macrophages, and dendritic cells, instead of MDSC. Myeloid APC can then go on to re-activate anergized memory T cells to re-gain their anti-tumor cytotoxic potential and promotes CTC apoptosis. Anti-tumor macrophages and disruption of CTC-M-FB cluster formation can also directly affect CTC proliferation, mobilization, and apoptosis, thus hindering their survival in the portal blood and preventing their metastatic progression.

that the epithelial to mesenchymal transition phenotype of CTC may be indicative of further development of aggressive CTC subpopulations with high metastatic potential [9, 42–47]. Using our FACS selection panels of CD44+CD147+EPCAM+/CK+CD45- for portal blood CTC in this and previous studies of PDAC CTC [7, 11] we have collected both EPCAM+ and CK+ populations of CTC whenever possible but have routinely not examined the characteristics of EPCAM+CK- and EPCAM-CK+ subpopulations within the portal blood. The delineation of these 2 subpopulations may be of use in future studies of the metastatic potential of PDAC CTC subpopulations as described by Li and Stanger [9]. The CD44+CD147+EPCAM +/CK+CD45- selection criteria is a useful tool for CTC isolation but could be less than 100% specific considering the noted variability in CTC biomarker expression seen in our analyses. Addition of more selective markers as they are discovered could allow for better definition of the metastatic potential of individual CTC clones within the portal blood population. Interstitial biomarkers such as Twist and Vimentin have been found advantageous to detection of aggressive CTC in lung [45, 47], breast [45], and recently, pancreatic cancers [42–46] in fluorescence immunohistochemical (IHC) staining and microscopy-based selection methods. Intracellular staining for vimentin has been suggested for distinguishing tumor cells from fibroblasts by IHC or flow cytometry. [44] Since stable intracellular staining for flow cytometric and IHC analyses requires fixation of cells, we were unable to find a workable antibody conjugate reagent and permeabilization method to detect such interstitial biomarkers to incorporate into our live CTC FACS selection panel that would maintain viability of CTC for the *ex vivo* culture analyses used in these current studies. Development of such biomarkers for future viable cell studies could be a beneficial addition to further define the PDAC CTC subpopulations most likely to migrate out of the portal blood and metamorphosize into metastatic seeding clones [15, 42–47].

Targeting CTC-myeloid cell interactions presents a favorable target for intervention (intraoperatively and/or immediate post-operatively) directed at the portal circulatory reservoir of residual CTC among patients undergoing surgical treatment for PDAC. A working model for such targeted interventions might employ portal vein infusion of combination immunotherapy aimed at inhibiting CXCR2/IL-8 and M-CSF-IL34/CSF1R signaling driven by PDAC CTC, thus allowing for myeloid differentiation away from the influence of CTC so that myeloid APC differentiation predominates over MDSC/M-FB development (**Fig 9B**).

Studies of IL-34 production by tumor cells in other cancer types indicates that aberrant expression of this growth factor cytokine can directly drive immune suppressive myeloid cells in their role in tumor resistance to T-cell directed checkpoint therapies. Blocking IL-34 binding to CSF1R may improve the efficacy of checkpoint blockade immunotherapy [36, 39, 48, 49]. Our findings also indicate that IL-34 is being produced by PDAC CTC in the portal blood and may have a profound effect on skewing myeloid differentiation toward immunosuppression and maintenance of anergy in tumor-directed T cells. Given the high abundance of such suppressor cells in the portal blood [7] and PDAC resistance to checkpoint therapies [26, 39, 49], our findings indicate an intersection of tumor cell effects on myeloid differentiation signaling pathways with PDAC immunoresistance as a mechanism of action for combined immunotherapies including anti-IL-8/anti-CSF1R based therapies such as Cabiralizumab and

BMS-986253 to prevent liver metastasis contributions to PDAC progression after surgical resection. By their direct disruption of CTC-M-FB cluster formation, combined anti-CSF1R/ anti-IL-8 therapies can also directly affect CTC proliferation, mobilization, and apoptosis; and thereby, hinder CTC survival in the portal blood and preventing their metastatic progression. In contrast to T cell-directed checkpoint therapies, myeloid-directed therapies can promote development of anti-tumor myeloid-derived APC, which when activated can downregulate immune checkpoint signals while promoting signaling to re-activate anergic anti-tumor T cells found in the portal blood to directly attack PDAC CTC.

## Summary conclusions

Circulating tumor cells (CTC) shed from primary PDAC tumors interact with myeloid cells in the portal blood and have the potential to sway myeloid differentiation toward immunosuppressive MDSC and tumor supportive myeloid-derived fibroblasts (M-FB) through activation of CSF1R and CXCR2 signaling pathways mediated by M-CSF/IL-34 and IL-8, respectively. CTC can also use autocrine CSF1R signaling to enhance their metastatic potential. Therapeutic antibody blockade of CSF1R and IL-8 signaling may disrupt CTC interactions with myeloid cells, impairing CTC proliferation and motility, while allowing for GM-CSF-driven APC to re-activate tumor responsive T lymphocytes that had been anergized by immunosuppressive environment of PDAC portal blood.

## Supporting information

**S1 Fig. Comparison of CTC numbers *ex vivo* and after *ex vivo* cluster cultures in pre-surgically treated patient samples to samples from patients without pre-surgical treatment. A)** CTC were aseptically isolated from freshly ficoll gradient separated, uncultured portal blood mononuclear cells (PortalBMC) using high-speed FACS. CD44+CD147+EPCAM+/CK +CD45- CTC numbers were compared between 20 PDAC patients that received pre-operative treatment (Neo adjuvant chemotherapy ± radiation) and 8 patients who had no pre-operative therapy. As previously reported [7, 11], PDAC patients receiving neo adjuvant chemotherapy with or without radiation had lower portal blood CTC counts than untreated patients (p = 0.0361; with noted variability in sample population distribution). Sample values are shown by open circles, horizontal bars indicate mean of values and error bars depict standard error of the mean. **B)** Comparison of PDAC patient portal CTC numbers in uncultured PortalBMC sub-grouped by tumor stages (T1-3) with (N+) or without (N0) lymph node involvement. TN staging was derived from the post-operative resected tumor tissue pathological review for PDAC patients verified after analysis of portal blood samples from the same patients. PDAC Patients receiving pre-operative treatments are depicted by white circles. PDAC patients without neoadjuvant treatment are represented by black dots. Horizontal bar indicates mean of values and error bars depict standard error of the mean. No statistical difference in portal blood CTC counts was seen between tumor stage with or without lymph node involvement. **C)** A correlation analysis of the 8 PDAC patients without neoadjuvant therapy showed a weakly linear but statistically significant negative relationship between portal blood CTC numbers and tumor (T) staging, with more CTC detected in lower stage tumors than in higher stage tumors (p = 0.0471; $r^2$ = 0.4662). **D)** Correlation analysis showed no relationship between portal blood CTC numbers and tumor (T) staging in 20 PDAC patients receiving neo adjuvant treatment. Two PDAC patients (one T1N0, one T3N0) in the study did not have pre-culture FACS isolation of CTC performed; and therefore, are not included in this set of analyses.
(TIF)

**S2 Fig. RNA expression analysis of single cell sorted PDAC individual CTC clones. A)** FACS isolated single clones (SCS) of CTC from 2 PDAC patients' uncultured PortalBMC were collected directly into RNALater and shipped frozen to GeneWiz for RNA extraction and analysis using RNA Microsequencing (RNA SEQ, GeneWiz, NJ,USA). Analysis of the top 100 genes expressed by individual clonal CTC (SCS) and oligoclonal CTC FACS isolations (OLIGO) for each patient were compared. **A)** Heatmaps depict the top 30 genes found by GeneWiz analyses for 3 SCS clones and 1 sample of oligo CTC isolates for the 2 patients as examples of variability among clones of each patient. Further bioinformatic analyses of the expression patterns within each patients' clonal variants revealed some clones with elevated expression of myeloid cell growth factor signaling genes and increased expression of the oncogene *RAN* and its regulatory target gene *SPP1*, which encodes for the tumor motility promoting protein osteopontin [S1], among the differentially expressed genes in PDAC clones. For confirmatory analyses of these candidate genes' expression, FACS-isolated single clones of CTC from 12 patients with PDAC were individually extracted and yielded limited RNA samples for quantitative RT-PCR analysis. GAPDH RNA expression in clonal cell isolates (data not shown) was used for RNA copy number calculations of *KRASmut*, oncogene *RET1* [S2], *CSF2*, *CSF2R*, and *CSF1R* expression per clone. **B)** Graph depicts results from analysis of CTC SCS from 12 PDAC patients using 3 CTC SCS clones per patient, confirming high expression of *KRASmut*, *RET1*, and *CSF1R* RNA with variability seen within each patient's clonal set. Two CTC clones from a single patient showed high expression of *CSF2R* as well. **C)** Quantitative RT-PCR analysis of RNA from SCS CTC clones of 5 individual PDAC patient portal blood samples showed high variability of *RAN* oncogene expression while its substrate target gene, *SPP1*, remained relatively stable in its expression across clones. Sample values are shown by open circles, horizontal bar indicates mean of values and error bars depict standard error of the mean. References for S2 Fig. S1. Saxena S, Gandhi A, Lim P-W, Relles D, Sarosiek K, Kang C, et al. RAN GTPase and Osteopontin in Pancreatic Cancer. Pancreat Disord Ther. 2014; 3 (1): 113–132. S2. Reza J, Almodovar AJO, Srivastava M, Veldhuis PP, Patel S, Fanaian N, et al. K-Ras Mutant Gene Found in Pancreatic Juice Activated Chromatin From Peri-ampullary Adenocarcinomas. Epigenetic Insights. 2019; 12: 1–8.
(TIF)

**S3 Fig. Comparison of proliferation of FACS isolated cells in defined *ex vivo* cluster cultures in pre-surgically treated patient samples to samples from patients without pre-surgical treatment.** Data depicted are derived from **Fig 1** panel E. Among the 16 PDAC patient samples depicted, 13 patients received neoadjuvant treatment with chemotherapy with or without radiation therapy prior to resection. The remaining 3 patients that did not undergo neoadjuvant therapy had higher CTC counts overall after *ex vivo* culture, but otherwise changes in CTC proliferation in the defined isolated cell *ex vivo* cultures were not significantly different from that seen in those of patients receiving neoadjuvant treatment. Bars indicate mean of values and error bars depict standard error of the mean.
(TIF)

**S4 Fig. Comparison of cytokine and growth factor levels in portal and peripheral blood of pre-surgically treated patient plasma samples to samples from patients without pre-surgical treatment.** Data depicted are derived from **Fig 2** Panel C. ELISA (Boster Scientific) analysis of matched portal (white bars) and peripheral (grey bars) plasma of 19 patients with adequate volume for extended analysis, with 16 who received neo adjuvant therapy (chemotherapy ± radiation therapy) prior to resection were compared for concentrations of PGE2, GM-CSF, M-CSF, IL-34, IL-6, IL-8, IFNγ, IL1α, and IL1β. (**A**) The three patients in this

data set that did not receive neoadjuvant had high levels of M-CSF, IL-6, and IL1β in their portal blood plasma but little or no detected in their peripheral blood plasma. With so few untreated patient samples represented, no reliable statistical analysis was possible for comparison. For the 16 patients with neoadjuvant treatment (**B**), statistical differences were seen in increased M-CSF, IL-6, and IL1β concentrations in portal blood compared with peripheral blood plasma. Red lines indicate literature values for concentrations in non-malignant human plasma; p values in red indicate significant differences found in PDAC portal and peripheral blood plasma samples used in this study compared to control values cited in the literature. Sample bar indicates mean of values and error bars depict standard error of the mean, with * indicating $p < 0.05$, ** $p < 0.01$, ns = no significant difference seen, and *** indicate where samples with undetectable analyte levels precluded statistical comparison. (TIF)

**S1 File.**
(HTML)

**S2 File.**
(HTML)

**S1 Data.**
(XLSX)

## Acknowledgments

Special thanks to L Boroughs, PhD, and B Nepon-Sixt, PhD of Bristol Myers Squibb Research Division, Tampa and A Khaled, PhD of University of Central Florida for intellectual review and discussions of the study.

## Author Contributions

**Conceptualization:** Juan Pablo Arnoletti, Sally A. Litherland.

**Data curation:** Joseph Reza, Armando Rosales, Alberto Monreal, Na'im Fanaian, Suzanne Whisner, Milan Srivastava, Julia Rivera-Otero, Quang Tran, Sally A. Litherland.

**Formal analysis:** Juan Pablo Arnoletti, Joseph Reza, Na'im Fanaian, Gongxin Yu, Quang Tran, Sally A. Litherland.

**Funding acquisition:** Juan Pablo Arnoletti, Otto Phanstiel IV, Sally A. Litherland.

**Investigation:** Juan Pablo Arnoletti, Armando Rosales, Alberto Monreal, Na'im Fanaian, Suzanne Whisner, Julia Rivera-Otero, Gongxin Yu, Otto Phanstiel IV, Deborah A. Altomare, Sally A. Litherland.

**Methodology:** Na'im Fanaian, Milan Srivastava, Sally A. Litherland.

**Project administration:** Juan Pablo Arnoletti, Sally A. Litherland.

**Resources:** Juan Pablo Arnoletti, Joseph Reza, Armando Rosales, Milan Srivastava, Otto Phanstiel IV, Deborah A. Altomare, Sally A. Litherland.

**Supervision:** Juan Pablo Arnoletti, Sally A. Litherland.

**Validation:** Juan Pablo Arnoletti, Na'im Fanaian, Suzanne Whisner, Julia Rivera-Otero, Gongxin Yu, Sally A. Litherland.

**Visualization:** Sally A. Litherland.

**Writing – original draft:** Sally A. Litherland.

**Writing – review & editing:** Juan Pablo Arnoletti, Joseph Reza, Armando Rosales, Alberto Monreal, Na'im Fanaian, Suzanne Whisner, Milan Srivastava, Julia Rivera-Otero, Gongxin Yu, Otto Phanstiel IV, Deborah A. Altomare, Quang Tran, Sally A. Litherland.

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
