## [Decision Letter · Decision Letter 0]

13 Dec 2021

PONE-D-21-29413Pancreatic Ductal Adenocarcinoma (PDAC) Circulating Tumor Cells Influence Myeloid Cell Differentiation to Support their Survival and Immunoresistance in Portal Vein CirculationPLOS ONE

Dear Dr. Litherland,

Thank you for submitting your manuscript to PLOS ONE. After careful consideration, we feel that it has merit but does not fully meet PLOS ONE’s publication criteria as it currently stands. Therefore, we invite you to submit a revised version of the manuscript that addresses the points raised during the review process.==

We look forward to receiving your revised manuscript.

Kind regards,

Dominique Heymann, Ph.D.

Academic Editor

PLOS ONE

Journal Requirements:

I have read the journal's policy and the authors of this manuscript have the following competing interests:

SAL, JPA, NF, SW, JRO, AR, and MS receive humanized antibodies and funding for this study from a grant from Bristol Myers Squibb (Grant BMS CA025-016); no personal funds, research support only.

Reviewers' comments:

Reviewer's Responses to Questions

**Comments to the Author**

1. Is the manuscript technically sound, and do the data support the conclusions?

Reviewer #1: Yes

Reviewer #2: Partly

2. Has the statistical analysis been performed appropriately and rigorously? 

Reviewer #1: Yes

Reviewer #2: Yes

3. Have the authors made all data underlying the findings in their manuscript fully available?

Reviewer #1: Yes

Reviewer #2: No

4. Is the manuscript presented in an intelligible fashion and written in standard English?

Reviewer #1: Yes

Reviewer #2: No

5. Review Comments to the Author

Reviewer #1: Thank you for inviting me to evaluate the article titled“Pancreatic Ductal Adenocarcinoma (PDAC) Circulating Tumor Cells Influence Myeloid Cell Differentiation to Support their Survival and Immunoresistance in Portal Vein Circulation.” In this paper, the authors want to show that Circulating tumor cells (CTC) shed from primary PDAC tumors interact with myeloid cells in the portal blood and can sway myeloid differentiation toward immunosuppressive MDSC and tumor supportive myeloid derived fibroblasts (M-FB) through activation of CSF1R and CXCR2 signaling pathways mediated by M-CSF/IL-34 and IL-8, respectively. However, there are some problems in expression and defects in the results of the experiment.

The author used CD147、CD47、EPCAM and Cytokeratin to characterize circulating tumor cells, but according to I know, high expression interstitial marker CTC is more likely to be transferred. Why does the author do not add an interstitial marker?

If the above problem can be solved, this article is worth receiving.

Reviewer #2: Summary of the research and overall impression

This work is focused on the study of the circulating tumor cells isolated from the portal blood. Most studies have focused on CTC detection and counting in peripheral blood samples but fewer reports have focused in capturing and detecting CTCs in vessels closer to the tumor, especially in the main veins that drain blood from the organ invaded by the cancer (‘close-to-the-tumor liquid biopsy’). I consider this a very positive aspect of this approach. In this study portal vein CTCs were cultured ex vivo together with myeloids cells to study proliferation, motility and cluster formation compared to CTC alone. The interaction between these two cell types causes myeloid cells to differentiate into an immunosuppressive phenotype mediated by the release of M-CFS/IL-34 and IL-8.

Blocking this interaction would prevent this phenotype change in myeloid cells and is postulated as a possible treatment for pancreatic cancer, which is one of the most aggressive solid tumors for which there is little chance of success with current treatments.

The purpose of the research is important and of interest to potential readers, however, I have some doubts and problems in understanding the tests performed. My biggest doubt is related to the number of samples used in each assay, I don't quite understand the diversity in the samples used, especially in assays that do not depend on the existence of CTCs in the patient's blood. And also, the fact of assuming that BMC are M-FB, when in fact they are a heterogeneous mixture of different cell types.

The text is a bit difficult to follow as there are no clearly delimited sections, for example with a subtitle. The images in general could be improved in relation to the quality of the microscopy images and the size of the axes of the graphs which is sometimes difficult to see. Some images are poorly explained in the text.

Material and Methods:

CTC isolation method is based on the expression of epithelial cell markers, such as CK19 or EpCAM, couldn´t you have missed CTCs undergoing EMT?

Sample collection: The description of the patients indicates that 18 patients had received chemotherapy prior to surgery (Folfirinox or Gemcitabine/Abraxane) and 8 of them had even received radiation. It is described that the treatments will influence the release and the characteristics of CTC into the blood. This factor could be introducing noise in the results. I understand that the number of patients is low but it would be convenient for the authors to analyze the results separately for patients who have received therapy and those who have not. This may even give information on whether therapy affects the interaction of myeloid cells with CTCs.

I understand that patients will be in stage I or II disease if they are amenable to surgery but it might be useful to make this clear in the text. Inclusion criteria could be better described.

FACS: I have not checked all the antibody references but for example, for CD44 the references indicated correspond to CD34 and CD81 (line 110). Likewise, I don't know if the authors meant to refer to CD47 since CD44 is not among the markers used for cytometry of CTCs or M-FB.

RNA: Any assessment of RNA quality?

Results:

In Figure 1, I would suggest rearranging the panels in the order in which they are cited in the text. It is a bit odd to have to look at figure 1c first. This should be 1a if it is the first one being discussed.

Figures 1C-H how have they been taken? Why do we know they are CTC or M-FB? There are supposed to be two dyes but the image quality does not allow to see anything.

I find figure 1a rather incomprehensible. The authors claim that CTC-MFB co-culture induces CTC proliferation. I don't understand this result in the graph. I don't understand this result in the graph.

Of the 16 patients used in this trial, how many had received therapy? Does that affect the outcome?

Figure 2A is not referred in the text. I understand that for trials with CTCs the number of patients is not the total, but to analyze the levels of cytokines and growth factors in blood, why not all patients? What are the red stripes in figure 2A? Are they normal reference values? A bit unusual way to put them.

According to the text: “GM-CSF levels in PDAC patient portal blood plasma were highly elevated compared to normal peripheral blood (p=0.0305) while M-CSF levels were moderately elevated over 247 normal (p=0.0383)” Why do you compare portal vein levels with normal peripheral blood levels? You would have to compare portal blood GM-CSF levels with normal levels there and peripheral blood levels with normal levels there. Or are they the same?

Figure 2c would benefit from a legend stating that the white bars are portal blood and the gray bars are peripheral blood. Significant differences are marked between the two blood types but are they different from the reference levels in each compartment? Now, however, 24 patients are used for this figure. The criteria for this should be explained.

Here as before, treatments may have influenced the release of these factors into the blood.

Figure 3: The lettering on panels B-E is virtually unnoticeable.

One doubt that arises, perhaps because I have not understood it well, is that the authors refer to M-FB populations and blood mononuclear cells (BMC) indistinctly. But when isolated by FICOL BMC, this is a heterogeneous population of cells. In the test corresponding to Figure 3, they take BMC to study the interaction between CTC-MFB and in reality, it cannot be assumed that what is there is only M-FB.

Figure 3F is not clearly visible. Perhaps it could be divided into several graphs, one for each chemokine. Similarly, if all comparisons are made with the first graph, this can be indicated in the figure caption and there is no need to put so many dashes above the figure, you can put the asterisks indicating statistical significance only above each bar. This applies to other figures in the manuscript.

Figure 3M and 3N are vaguely explained in the figure legend. Annexin V plot should be supplemented.

Figure 4 is not explained in the text. It is very difficult to understand because it makes a statement and refers to the entire figure, panels A to D, without further explanation of what the different images show. What does zero mean, in the second bar of the graphs? For me, figure 4 does not contribute anything because I do not understand it.

Figure 5 is not explained either. This causes the reader's attention to be lost because it is very difficult to draw conclusions when they do not explain what they see in the graphs. It requires extra effort on the part of the reader.

Figure 6 focuses on gene expression changes in CTCs and M-FBs. In the materials and methods, the whole process of sequencing is described but in reality the only thing that is described is the expression of the same markers that were previously studied. The whole paragraph in materials and methods referring to the identification of differentially expressed genes does not make much sense with the results that are being presented in this article.

Figure 7A does not seem to have significant differences, but in the text the impression conveyed is that the treatment reduces the state of anergy (“treatment of PortalBMC cultures with anti-CSF1R, anti-IL-8, and/or anti-IL34 alone or in combination showed a 2-3-fold reduction of anergy among T cells). Perhaps this statement or the graph should be explained better.

Throughout the text there are a number of typos in the wording that could easily be corrected:

For example, in line 140 it says cells cells; 147 it says cluster clusters. Line 154 it does not have a verb (like were used)

Line 267, there is a dot after IL-8 that should be a colon.

6. PLOS authors have the option to publish the peer review history of their article (what does this mean?). If published, this will include your full peer review and any attached files.

Reviewer #1: No

Reviewer #2: No

---

## [Author Response · Author response to Decision Letter 0]

14 Feb 2022

Arnoletti et al-Response to Reviewers

We the authors sincerely thank the reviewers for their insightful comments for the improvement of our manuscript. We have taken their comments to heart and have done a thorough review and revision of the entire manuscript including updating all figures and adding 4 supplemental figures in the text here for our responses to answer the reviewers’ clarification requests. We leave it to the reviewers and editorial board to decide if any of these supplemental figures should be included in the final publication. Again, thank you for supporting our efforts to present our data accurately and clearly and improving the manuscript by your review. 

Below, we re-print the review comments sent to us in italics and answer the reviewers’ queries and comments point by point (shown in indented regular text paragraphs below each comment and in the supplemental figures depicted within these responses). 

“Reviewer #1: Thank you for inviting me to evaluate the article titled “Pancreatic Ductal Adenocarcinoma (PDAC) Circulating Tumor Cells Influence Myeloid Cell Differentiation to Support their Survival and Immunoresistance in Portal Vein Circulation.” In this paper, the authors want to show that Circulating tumor cells (CTC) shed from primary PDAC tumors interact with myeloid cells in the portal blood and can sway myeloid differentiation toward immunosuppressive MDSC and tumor supportive myeloid derived fibroblasts (M-FB) through activation of CSF1R and CXCR2 signaling pathways mediated by M-CSF/IL-34 and IL-8, respectively. However, there are some problems in expression and defects in the results of the experiment.”

We thank the reviewer for their kind comments. We agree that the original manuscript as uploaded had defects in presentation and organization that we have hopefully rectified in our thorough review of the entire document, updating data analyses, and revision of the text and figures.

“The author used CD147、CD47、EPCAM and Cytokeratin to characterize circulating tumor cells, but according to I know, high expression interstitial marker CTC is more likely to be transferred. Why does the author do not add an interstitial marker? If the above problem can be solved, this article is worth receiving.”

We apologize for the typographical error in the Methods section of the manuscript (now corrected on line 131 in the revised manuscript) that misrepresented the biomarker signature we used in flow cytometric and FACS analyses used in the study. The correct profile used for positive detection and FACS isolation of CTC was CD44+CD147+EPCAM+/CK+. We have added references 22 & 23 describing the use of the CD147 and CD44 biomarkers. We agree with the reviewer that use of interstitial biomarkers such as Twist or Vimentin would be useful to identify potential CTC clonal subsets more specifically if possible. However, as seen in published studies [references added: 43-47] using these biomarkers rely on fixed, permeabilized cell fluorescent analyses (IHC fluorescence microscopy and intracellular flow cytometry), which is not possible to use in our studies requiring viable patient cell isolation for ex vivo culture growth to test responses to myeloid cell interactions and live cell signal transduction in the presence of inhibitory antibody treatments. We acknowledge and address this limitation in the Discussion section of the revised manuscript (lines 742-755).

“Reviewer #2: Summary of the research and overall impression

This work is focused on the study of the circulating tumor cells isolated from the portal blood. Most studies have focused on CTC detection and counting in peripheral blood samples but fewer reports have focused in capturing and detecting CTCs in vessels closer to the tumor, especially in the main veins that drain blood from the organ invaded by the cancer (‘close-to-the-tumor liquid biopsy’). I consider this a very positive aspect of this approach. In this study portal vein CTCs were cultured ex vivo together with myeloid cells to study proliferation, motility and cluster formation compared to CTC alone. The interaction between these two cell types causes myeloid cells to differentiate into an immunosuppressive phenotype mediated by the release of M-CFS/IL-34 and IL-8. Blocking this interaction would prevent this phenotype change in myeloid cells and is postulated as a possible treatment for pancreatic cancer, which is one of the most aggressive solid tumors for which there is little chance of success with current treatments.

The purpose of the research is important and of interest to potential readers, however, I have some doubts and problems in understanding the tests performed. My biggest doubt is related to the number of samples used in each assay, I don't quite understand the diversity in the samples used, especially in assays that do not depend on the existence of CTCs in the patient's blood.” 

We truly appreciate the reviewer’s grasp of our findings despite the presentation errors and need for clarification found. We have done an end-to-end review of the manuscript to re-organize it and endeavored to clarify the points in the data that the reviewer observed.

We agree that the variations in sample numbers in each analyses was not ideal though we strived to maximize the number of analyses preformed on each sample. However, the inherent variation in patient samples and limitations of collecting and working with viable, minimally processed CTC samples from surgical patients preclude the use of every sample in all analyses. Patient samples were collected during surgery and analyzed blind prior to final pathological analyses and chart review, so we were unable to select samples based on staging or prior treatment. Samples were processed as close to collection as possible so that minimal time/manipulation could affect CTC biology prior to ex vivo analyses, with the clinical/surgical treatment of patients was prioritized above research sample collection in all instances. Patient sample variations in plasma quality/available volume, CTC collectable numbers, and technical limitations sometimes did not allow for equal plasma or cell samples to be available from each participant for this study. Analyses dependent on uncultured FACS isolation of CTC (such as those described in Figures 1 & 6 of the revised manuscript and below in Supplemental Figures 1&2) were limited in available viable cells and extracted RNA for these analyses as well as the available PortalBMC sample remaining for subsequent culture-based analyses. In addition, two patients’ surgery scheduling and technical limitations precluded pre-culture assessment of CTC/MFB from PortalBMC so that their samples could not be included in these data sets. Financial constraints limited the number of samples that could be analyzed by BioLegend Legendplex analysis and RNA SEQ. Portal plasma samples were available from all 30 patients but available volume and reagent costs limited their use in soluble factor (Figure 2) and U937 conditioned media studies (Figure 4).

 

“And also, the fact of assuming that BMC are M-FB, when in fact they are a heterogeneous mixture of different cell types.”

We thank the reviewer for pointing out the need to clarify these terms in the manuscript and to better describe which analyses were done using CTC and M-FB FACS isolated prior to culture (protocol previously described in Reference 11 and depicted in Figures 1 and 6 in the revised manuscript and Supplemental Figures 1&2 in the text below) and which other analyses were done using unsorted portal blood mononuclear cell (PortalBMC) in ex vivo cultures with CTC and M-FB subpopulations characterized and isolated after culture (Figures 3, 5, 7 & 8 in the revised manuscript). We have added more detail throughout the revised manuscript text and in all the Figure legends to more clearly delineate the distinction between when uncultured FACS-isolated CTC and M-FB were used and when unsorted PortalBMC were used for the analyses.

Supplemental Fig. 1 (found in Supplemental Information uploaded files). Comparison of CTC Numbers ex vivo and after ex vivo Cluster Cultures in Pre-Surgically Treated Patient Samples to Samples from Patients without Pre-Surgical Treatment. A) CTC were aseptically isolated from freshly ficoll gradient separated, uncultured portal blood mononuclear cells (PortalBMC) using high-speed FACS. CD44+CD147+EPCAM+/CK+CD45- CTC numbers were compared between 20 PDAC patients that received pre-operative treatment (Neo adjuvant chemotherapy ± radiation) and 8 patients who had no pre-operative therapy. As previously reported [7,11], PDAC patients receiving neo adjuvant chemotherapy with or without radiation had lower portal blood CTC counts than untreated patients (p=0.0361; with noted variability in sample population distribution). Sample values are shown by open circles, horizontal bars indicate mean of values and error bars depict standard error of the mean. B) Comparison of PDAC patient portal CTC numbers in uncultured PortalBMC sub-grouped by tumor stages (T1-3) with (N+) or without (N0) lymph node involvement. TN staging was derived from the post-operative resected tumor tissue pathological review for PDAC patients verified after analysis of portal blood samples from the same patients. PDAC Patients receiving pre-operative treatments are depicted by white circles. PDAC patients without neoadjuvant treatment are represented by black dots. Horizontal bar indicates mean of values and error bars depict standard error of the mean. No statistical difference in portal blood CTC counts was seen between tumor stage with or without lymph node involvement. C) A correlation analysis of the 8 PDAC patients without neoadjuvant therapy showed a weakly linear but statistically significant negative relationship between portal blood CTC numbers and tumor (T) staging, with more CTC detected in lower stage tumors than in higher stage tumors (p=0.0471; r2=0.4662). D) Correlation analysis showed no relationship between portal blood CTC numbers and tumor (T) staging in 20 PDAC patients receiving neo adjuvant treatment. Two PDAC patients (one T1N0, one T3N0) in the study did not have pre-culture FACS isolation of CTC performed; and therefore, are not included in this set of analyses. 

Supplemental Fig. 2 (found in Supplemental Information uploaded files). RNA Expression Analysis of Single Cell Sorted PDAC individual CTC clones. A) FACS isolated single clones (SCS) of CTC from 2 PDAC patients’ uncultured PortalBMC were collected directly into RNALater and shipped frozen to GeneWiz for RNA extraction and analysis using RNA Microsequencing (RNA SEQ, GeneWiz, NJ,USA). Analysis of the top 100 genes expressed by individual clonal CTC (SCS) and oligoclonal CTC FACS isolations (OLIGO) for each patient were compared. A) Heatmaps depict the top 30 genes found by GeneWiz analyses for 3 SCS clones and 1 sample of oligo CTC isolates for the 2 patients as examples of variability among clones of each patient. Further bioinformatic analyses of the expression patterns within each patients’ clonal variants revealed some clones with elevated expression of myeloid cell growth factor signaling genes and increased expression of the oncogene RAN and its regulatory target gene SPP1, which encodes for the tumor motility promoting protein osteopontin [21], among the differentially expressed genes in PDAC clones. For confirmatory analyses of these candidate genes’ expression, FACS-isolated single clones of CTC from 12 patients with PDAC were individually extracted and yielded limited RNA samples for quantitative RT-PCR analysis. GAPDH RNA expression in clonal cell isolates (data not shown) was used for RNA copy number calculations of KRASmut, oncogene RET1, CSF2, CSF2R, and CSF1R expression per clone. B) Graph depicts results from analysis of CTC SCS from 12 PDAC patients using 3 CTC SCS clones per patient, confirming high expression of KRASmut, RET1, and CSF1R RNA with variability seen within each patient’s clonal set. Two CTC clones from a single patient showed high expression of CSF2R as well. C) Quantitative RT-PCR analysis of RNA from SCS CTC clones of 5 individual PDAC patient portal blood samples showed high variability of RAN oncogene expression while its substrate target gene, SPP1, remained relatively stable in its expression across clones. Sample values are shown by open circles, horizontal bar indicates mean of values and error bars depict standard error of the mean.

“The text is a bit difficult to follow as there are no clearly delimited sections, for example with a subtitle.”

We re-organized the text to hopefully improve its clarity and have incorporated subtitles as the reviewer has suggested.

“ The images in general could be improved in relation to the quality of the microscopy images and the size of the axes of the graphs which is sometimes difficult to see.”

Where possible, we have enlarged and worked to improve the images and graphic representations of our data in all the figures in the revised manuscript.

“ Some images are poorly explained in the text.”

We have added more detail in the text of the Figure legends on the images used in the figures of the revised manuscript.

“Material and Methods:

CTC isolation method is based on the expression of epithelial cell markers, such as CK19 or EpCAM, couldn´t you have missed CTCs undergoing EMT?”

The reviewer has made a valid point of the limitations on assessing EMT status of CTC in our studies. We have addressed this issue by clarifying the CTC biomarkers that were identified and how the CTC subpopulations were used in these studies in the Method section (lines 130-137), and how they could be used in future studies in the Discussion section (lines 736-740) in the revised manuscript.

“Sample collection: The description of the patients indicates that 18 patients had received chemotherapy prior to surgery (Folfirinox or Gemcitabine/Abraxane) and 8 of them had even received radiation. It is described that the treatments will influence the release and the characteristics of CTC into the blood. This factor could be introducing noise in the results. I understand that the number of patients is low but it would be convenient for the authors to analyze the results separately for patients who have received therapy and those who have not. This may even give information on whether therapy affects the interaction of myeloid cells with CTCs.”

We agree with the reviewer and have updated the proportions of neoadjuvant treated and untreated patients with new data added in the revised manuscript and re-analyzed the data of the study considering these groups. The results of these subgroup analyses are included in 4 Supplemental Figures 1-4 included here in the text of our responses. Beyond the increase in portal blood CTC numbers seen in uncultured (Supplemental Figure 1A, above) and cultured CTC similar to previously reported in references 7 &11, we did not see any change in statistical analyses in the data of these current studies by the breakout of untreated and neoadjuvant treated sample subpopulations. With further separation of chemotherapy type and radiation therapy subgroups, the data subsets were too small to accurately analyze differences between treatment types statistically. We have described the updated proportions of patient treated and untreated subgroups in the Methods section (lines 112-128). We defer to the reviewers and editors of the journal to judge whether any of the supplemental figures depicted here need to be included with the final publication of the paper. 

Supplemental Fig. 3(found in Supplemental Information uploaded files). Comparison of Proliferation of FACS isolated cells in Defined ex vivo Cluster Cultures in Pre-Surgically Treated Patient Samples to Samples from Patients without Pre-Surgical Treatment. Data depicted are derived from Fig 1 panel E. Among the 16 PDAC patient samples depicted, 13 patients received neoadjuvant treatment with chemotherapy with or without radiation therapy prior to resection. The remaining 3 patients that did not undergo neoadjuvant therapy had higher CTC counts overall after ex vivo culture, but otherwise changes in CTC proliferation in the defined isolated cell ex vivo cultures were not significantly different from that seen in those of patients receiving neoadjuvant treatment. Bars indicate mean of values and error bars depict standard error of the mean.

Supplemental Fig. 4(found in Supplemental Information uploaded files). Comparison of Cytokine and Growth Factor Levels in Portal and Peripheral Blood of Pre-Surgically Treated Patient Plasma Samples to Samples from Patients without Pre-Surgical Treatment. Data depicted are derived from Fig. 2 Panel C. ELISA (Boster Scientific) analysis of matched portal (white bars) and peripheral (grey bars) plasma of 19 patients with adequate volume for extended analysis, with 16 who received neo adjuvant therapy (chemotherapy ± radiation therapy) prior to resection were compared for concentrations of PGE2, GM-CSF, M-CSF, IL-34, IL-6, IL-8, IFNγ, IL1α, and IL1β. (A) The three patients in this data set that did not receive neoadjuvant had high levels of M-CSF, IL-6, and IL1β in their portal blood plasma but little or no detected in their peripheral blood plasma. With so few untreated patient samples represented, no reliable statistical analysis was possible for comparison. For the 16 patients with neoadjuvant treatment (B), statistical differences were seen in increased M-CSF, IL-6, and IL1β concentrations in portal blood compared with peripheral blood plasma. Red lines indicate literature values for concentrations in non-malignant human plasma; p values in red indicate significant differences found in PDAC portal and peripheral blood plasma samples used in this study compared to control values cited in the literature. Sample bar indicates mean of values and error bars depict standard error of the mean, with * indicating p<0.05, ** p<0.01, ns = no significant difference seen, and *** indicate where samples with undetectable analyte levels precluded statistical comparison. 

“I understand that patients will be in stage I or II disease if they are amenable to surgery but it might be useful to make this clear in the text. Inclusion criteria could be better described.”

The tumor stage information (lines 120-128) and inclusion criteria (lines 116-117) for the patients participating in the study have been added in the Methods Section. Analysis of uncultured portal blood CTC numbers considering final pathological diagnosis of tumor stage is included in Supplemental Figure 1B-D shown above in the response text. No statistically significant differences were seen between patients sub-grouped by TN staging (Supplemental Figure 1B, above). A weakly linear statistically significant correlation of CTC numbers with tumor T staging was seen in the patient subpopulation that did not receive pre-surgical chemotherapy ± radiation therapy (Supplemental Figure 1C, above); however, no comparable correlation was seen in the neoadjuvant treated patient subpopulation (Supplemental Figure 1D, above). These tumor stage comparison results are summarized in the Methods section lines 125-128 in the revised manuscript.

“FACS: I have not checked all the antibody references but for example, for CD44 the references indicated correspond to CD34 and CD81 (line 110).”

We thank the reviewer for finding this source error (BioLegend not BD Sciences, corrected in revised Methods section line 140) in the submitted manuscript and we have followed their advice to review all of antibody references mentioned in the paper. We have updated all references in the revision to reflect our most up-to-date sources for the reagents used in the studies.

“Likewise, I don't know if the authors meant to refer to CD47 since CD44 is not among the markers used for cytometry of CTCs or M-FB.”

We apologize for the typographical error in the Methods Section that erroneously referred to the biomarker CD47. The correct antibody reagent used was anti-CD44. The correct CTC identification biomarker panel has been listed on line 131 in the Methods section of the revised manuscript.

“RNA: Any assessment of RNA quality?”

The assessment of 260/280 ratio was used in evaluating RNA quality and quantity in the samples used in the study. This has been annotated in the revised manuscript Methods section lines 268-270.

“Results:

In Figure 1, I would suggest rearranging the panels in the order in which they are cited in the text. It is a bit odd to have to look at figure 1c first. This should be 1a if it is the first one being discussed.”

We agree and have rearranged Figure 1 as the reviewer has suggested in the revision of the manuscript.

“Figures 1C-H how have they been taken? Why do we know they are CTC or M-FB? There are supposed to be two dyes but the image quality does not allow to see anything.”

We have re-arranged the images in Figure 1 and enlarged them from the original images to hopefully improve their clarity. We have also added more detail on how the images were taken in the Figure 1 Legend (Results lines 334-361) and added more detailed explanation of the methodology used in these analyses and their limitations in their interpretation in the Methods section (lines 199-209) and Results section (lines 320-332). The identification of CTC and M-FB was based on the FACS-isolation biomarker panel used in their separation from PortalBMC prior to defined cell co-cultures (Methods lines 131-146 & 163-176, and reference 11).

“I find figure 1a rather incomprehensible. The authors claim that CTC-MFB co-culture induces CTC proliferation. I don't understand this result in the graph.”

We thank the reviewer for letting us know more detail on the methods used in these studies was needed to help with their presentation. In the re-organization of Figure 1 suggested by the reviewer, this figure panel is now 1E in the revised manuscript. We agree with the reviewer that the findings can be interpreted better as CTC/M-FB interactions enhancing or supporting CTC proliferation rather than inducing it. We have added more detail information on the pre-culture FACS isolation used in these defined cell type co-culture studies in the Methods section (lines 130-146 & 163-176, also detailed in reference 11) and in the Results section and Figure 1 Legend (Results lines 334-361). Based on statistical analysis, CTC numbers in cultures of CTC and M-FB co-cultured after FACS isolation were not statistically different than that seen in unsorted PortalBMC but were increased compared to FACS-isolated CTC cultured alone or added back to cultures of negatively selected cell populations out of PortalBMC lacking CTC and M-FB sorted populations (-CTC-MFB+CTC), suggesting the combination of CTC and M-FB cells is needed for cluster formation and enhanced CTC proliferation.

“Of the 16 patients used in this trial, how many had received therapy? Does that affect the outcome?”

Thirteen patients in this data set (Figure 1E now in the revised manuscript) had received pre-operative (neo adjuvant) treatment, 3 did not. Due to the low numbers of the untreated patients in the data set, no statistical analyses were possible to accurately compare the 2 subgroups. Graphs of the separated data from the 2 subgroups are depicted in Supplemental Figure 3 in response text above.

“Figure 2A is not referred in the text.”

We apologize for this typographical error. The Fig.2A was cited but not bolded as the other figure references were in the original manuscript text. We have corrected this in the revised manuscript (Results Section lines 369, 531, and 545).

“I understand that for trials with CTCs the number of patients is not the total, but to analyze the levels of cytokines and growth factors in blood, why not all patients?” 

Financial limitation prevented samples beyond the first 11 available from being analyzed by Legendplex multiple cytokine cytometry assay. Plasma samples were also used in U937 differentiation analyses and other studies not included in this report, leaving 24 out of 30 plasma samples available for analysis in ELISA (Figure 2B). The other 6 samples did not have adequate volumes available for analysis. After all of the above analyses, 19 patient samples had enough plasma available for matched peripheral and portal plasma analyses (Figure 2C) in expanded parallel ELISA analyses of M-CSF, IL-6, IL-1beta, GM-CSF, IL-34, IL-8, IL-1alpha, and IFNgamma.

“What are the red stripes in figure 2A? Are they normal reference values? A bit unusual way to put them.”

We appreciate the reviewer pointing out the need for clarification on the graph presentation. The horizontal red lines in the A, B, and C panel graphs of Figure 2 in the revised manuscript and in Supplemental Figure 4 depicted in response text above are indicators of published healthy control blood mean or median concentrations of the factors tested. The dotted vertical lines in Figure 2A indicate the standard deviation from the mean of these control levels given by the BioLegend Legendplex flow cytometry reagent kit. We have revised the Results text (lines 363-381) and Figure legends (Figure 2 lines 383-404) to better describe these indicators on the graphs. In the revision of the figures we have also corrected the positioning of these indicators that appeared to have shifted in the conversion of illustrations to eps and pdf formats required for upload submission.

“According to the text: “GM-CSF levels in PDAC patient portal blood plasma were highly elevated compared to normal peripheral blood (p=0.0305) while M-CSF levels were moderately elevated over normal (p=0.0383)” Why do you compare portal vein levels with normal peripheral blood levels? You would have to compare portal blood GM-CSF levels with normal levels there and peripheral blood levels with normal levels there. Or are they the same?”

We agree with the reviewer that direct comparison with healthy portal blood would have been better but because portal blood collection is only possible via surgical or invasive endoscopic intervention, we could not ethically collect healthy control portal blood from non-surgical patients or volunteers. The approved IRB protocols covering sample collections in this study limited portal or peripheral blood collection to pancreatic/GI oncological surgery patients. Therefore, we used published or assay manufacturer provided concentration levels for healthy peripheral blood comparisons in these analyses. We have included this explanation in the Methods (lines 244-250) in the revised manuscript.

“Figure 2c would benefit from a legend stating that the white bars are portal blood and the gray bars are peripheral blood.”

We have added the descriptions suggested by the reviewer to the Figure Legends of Figure 2 (Results lines 383-404) and Supplemental Figure 4 in the response text above.

“Significant differences are marked between the two blood types but are they different from the reference levels in each compartment?”

We have added the control level lines and statistically significant differences to the graphs in Figure 2C and Supplemental Figure 4 and included a description of these in their Figure Legends Figure 2 (Results lines 383-404) and Supplemental Figure 4 (in response text above).

“Now, however, 24 patients are used for this figure. The criteria for this should be explained.”

As mentioned above, the main limitation on samples used in these analyses was the volume of plasma available for matched peripheral and portal plasma sample comparisons. We have added an explanation of this limitation to the Figure Legends for Figure 2 (Results lines 383-404) and Supplemental Figure 4 (response text above).

 “Here as before, treatments may have influenced the release of these factors into the blood.”

We have included the requested analyses separating neo adjuvant treated and untreated patients in Supplemental Figure 4 (response text above). No significant changes in statistical analyses were seen though the limited number of untreated samples tested and the undetectable levels of some analytes precluded statistical analyses in the untreated patient sample set.

“Figure 3: The lettering on panels B-E is virtually unnoticeable.”

We agree with the reviewer that the uploaded pdf version of these images were not as clear as needed. We have re-done this Figure as recommended by the reviewer in the revised manuscript.

“One doubt that arises, perhaps because I have not understood it well, is that the authors refer to M-FB populations and blood mononuclear cells (BMC) indistinctly. But when isolated by FICOL BMC, this is a heterogeneous population of cells. In the test corresponding to Figure 3, they take BMC to study the interaction between CTC-MFB and in reality, it cannot be assumed that what is there is only M-FB.”

We thank the reviewer for bringing this clarity issue to our attention. We have revised the text of the Methods section (in vivo formed clusters lines 130-146, and defined cell cultures, lines 163-176, and unsorted PortalBMC cultures lines 199-209) and throughout the manuscript to include a better description of when CTC and M-FB were isolated by FACS prior to culture for defined co-culture experiments (as depicted in Figures 1&6 in the revised manuscript and Supplemental Figures 1 &2 in response text above), from when analyses were done with ficoll-separated PortalBMC first grown as a unsorted mixed culture and then CTC and M-FB were FACS isolated after culture for analyses (Figures 3, 5, 7 & 8). Both types of studies looked at CTC/M-FB interactions but the former looked at these in isolation; whereas, the latter analyzed them out of PortalBMC mixed cell culture.

“Figure 3F is not clearly visible. Perhaps it could be divided into several graphs, one for each chemokine.”

We have replaced the graph in Figure 3F with multiple graphs for each treatment series as requested by the reviewer and updated the Figure 3 Legend accordingly (Results lines 432-462) in the revision.

“ Similarly, if all comparisons are made with the first graph, this can be indicated in the figure caption and there is no need to put so many dashes above the figure, you can put the asterisks indicating statistical significance only above each bar. This applies to other figures in the manuscript.”

As the reviewer has suggested, we have simplified the notations on figures where possible in the revision. We have left brackets on some figures where comparisons are between specific groups within multiple ones depicted. 

“Figure 3M and 3N are vaguely explained in the figure legend. Annexin V plot should be supplemented.”

We have added more detail to the Legend of Figure 3 (Results lines 432-462) and Results text to better describe panels M & N (Results lines 427-430) and changed the axis title on the graph in panel M to denote that it depicts the AnnexinV binding data.

“Figure 4 is not explained in the text. It is very difficult to understand because it makes a statement and refers to the entire figure, panels A to D, without further explanation of what the different images show.”

We apologize for this oversight. A portion of the Methods section text describing the U937 cell line myeloid differentiation assays depicted in Figure 4 was inadvertently deleted in the upload of the original submission. It has been added back in the revision (Methods lines 252-263) and the Results section (lines 464-494). Figure 4 & its Legend (Results lines 496-519) have been revised to simplify the figure and include more detailed explanation of these results in its legend.

“ What does zero mean, in the second bar of the graphs?”

The ‘zero’ labels in Figure 4 denote U937 cultures that are supplemented with conditioned media from PortalBMC cultures that were not treated with any of the inhibitory antibodies tested. This label was used in Figure 4 to differentiate the conditioned media treated cultures from the control U937 culture grown without any conditioned media added as ‘no treatment (no trt)’ control for the assay. We have included a clearer definition of these labels in the revised Figure 4 Legend (Results lines 496-519).

“ For me, figure 4 does not contribute anything because I do not understand it.”

We hope that the revision of this figure and its legend (Results lines 496-519), additional explanatory text in the Results (lines 464-494), and the description of the U937 myeloid differentiation assay added back to the Methods (lines 252-263) helps the reviewer in their understanding of its relevance to the findings. These data show the suppression of U937 differentiation to all but M-FB when conditioned media from PortalBMC cultures treated with anti-IL-34 antibodies were used to supplement the U937 cultures. These results suggests that IL-34 in the portal PDAC PortalBMC cultures can act as a major component in the conditioned media promoting skewing myeloid differentiation away from APC and toward M-FB. 

“Figure 5 is not explained either. This causes the reader's attention to be lost because it is very difficult to draw conclusions when they do not explain what they see in the graphs. It requires extra effort on the part of the reader.”

We agree with the reviewer that this figure as originally presented was too complex to easily interpret and have removed some of the graphs depicting negative results in the revision and to make the data presentation less complicated. The purpose of presenting these data was to show the product level phenotype changes seen when cultures were treated with the inhibitory antibodies. We removed panels Figure 5 A-D in original manuscript that depicted the CSF1R and CXCR2 protein expression was unchanged on CTC and M-FB with the antibody treatments tested. We have now described these negative results in the Results text (lines 520-527). The new Figure 5 (Figure 5 E-H in the original manuscript) shows the marked changes in MCSF, IL-8, and IL-34 levels found in cultures supernatants with the treatments and shows that GM-CSF remains unchanged with the antibody treatments. We have revised the new Figure Legends to better detail these phenotypic changes observed with the inhibitory antibody treatments (Results lines 520-539 and Figure 5 Legend lines 541-554).

“Figure 6 focuses on gene expression changes in CTCs and M-FBs. In the materials and methods, the whole process of sequencing is described but in reality the only thing that is described is the expression of the same markers that were previously studied. The whole paragraph in materials and methods referring to the identification of differentially expressed genes does not make much sense with the results that are being presented in this article.”

We agree with the reviewer and have revised the presentation of these analyses and their results in the revision of the manuscript, including clarifying the RNA analyses in the Methods section (lines 148-161 & 265-283) and separating the data from uncultured, FACS-isolated CTC, in vivo formed CTC clusters and M-FB in new Figure 6 (formerly Figure 6A&B in original manuscript) and data from CTC and M-FB isolated from PortalBMC after culture in new Figure 7 (formerly Figure 6C-H in the original manuscript). We have added the reviewer requested data from RNA SEQ pilot analyses in Supplemental Figure 2 (in response text above). RNA SEQ data were used to look at intra-sample clonal variability which was apparent in pilot oligo clonal/SCS comparisons of 2 PDAC patients tested by RNA SEQ and in confirmatory RT-PCR tests with 12 patients’ SCS CTC RNA samples. The revised Legend for the Figure 6 is found in Results Section lines 570-582, and for the new Figure 7 in Results lines 602-621. We revised the Methods section to include an explanation of the findings of the RNA SEQ (Methods lines 148-161) and how they were used to design the RT-PCR studies depicted in Figures 6 & 7 in the revised manuscript. We have revised the text in the Results to be more conservative and accurate in our interpretation of these data (Results lines 556-645).

“Figure 7A does not seem to have significant differences, but in the text the impression conveyed is that the treatment reduces the state of anergy (“treatment of PortalBMC cultures with anti-CSF1R, anti-IL-8, and/or anti-IL34 alone or in combination showed a 2-3-fold reduction of anergy among T cells). Perhaps this statement or the graph should be explained better.”

The reviewer is correct, there are not enough data from this limited data set to draw any statistically significant conclusions, only to show a trend toward decreased T cell anergy with the treatments depicted. We have revised the text and Figure Legend (now new Figure 8) in the revised manuscript (Results text lines 647-659 and new Figure 8 Legend Results lines 669-690) to clarify this limited interpretation of the data.

“Throughout the text there are a number of typos in the wording that could easily be corrected:

For example, in line 140 it says cells cells; 147 it says cluster clusters. Line 154 it does not have a verb (like were used) Line 267, there is a dot after IL-8 that should be a colon.”

We apologize for the confusion these errors caused in the review. We have done an end-to-end editorial review of the manuscript and incorporated corrections to the items the reviewer listed as well as others found in the review process. We have also re-organized the manuscript to hopefully smooth the flow of the narrative throughout. We thank the reviewer for their detailed review comments that have helped us to improve the manuscript for re-submission.

---

## [Decision Letter · Decision Letter 1]

8 Mar 2022

Pancreatic Ductal Adenocarcinoma (PDAC) Circulating Tumor Cells Influence Myeloid Cell Differentiation to Support their Survival and Immunoresistance in Portal Vein Circulation

PONE-D-21-29413R1

Dear Dr. Litherland,

We’re pleased to inform you that your manuscript has been judged scientifically suitable for publication and will be formally accepted for publication once it meets all outstanding technical requirements.

Kind regards,

Dominique Heymann, Ph.D.

Academic Editor

PLOS ONE

Additional Editor Comments (optional):

Reviewers' comments:

Reviewer's Responses to Questions

**Comments to the Author**

1. If the authors have adequately addressed your comments raised in a previous round of review and you feel that this manuscript is now acceptable for publication, you may indicate that here to bypass the “Comments to the Author” section, enter your conflict of interest statement in the “Confidential to Editor” section, and submit your "Accept" recommendation.

Reviewer #2: All comments have been addressed

2. Is the manuscript technically sound, and do the data support the conclusions?

Reviewer #2: Yes

3. Has the statistical analysis been performed appropriately and rigorously? 

Reviewer #2: Yes

4. Have the authors made all data underlying the findings in their manuscript fully available?

Reviewer #2: Yes

5. Is the manuscript presented in an intelligible fashion and written in standard English?

Reviewer #2: Yes

6. Review Comments to the Author

Reviewer #2: The authors have answered all my questions in an exceptional manner. My opinion is that they have provided a clearer and better structured text and all the additional material (e.g. figures) has contributed to improve the quality of the data presented.

7. PLOS authors have the option to publish the peer review history of their article (what does this mean?). If published, this will include your full peer review and any attached files.

Reviewer #2: No

---

## [Editor Report · Acceptance letter]

12 Mar 2022

PONE-D-21-29413R1 

Pancreatic Ductal Adenocarcinoma (PDAC) Circulating Tumor Cells Influence Myeloid Cell Differentiation to Support their Survival and Immunoresistance in Portal Vein Circulation 

Dear Dr. Litherland:

I'm pleased to inform you that your manuscript has been deemed suitable for publication in PLOS ONE. Congratulations! Your manuscript is now with our production department. 

Kind regards, 

on behalf of

Pr. Dominique Heymann 

Academic Editor

PLOS ONE